# Cortical dendritic activity correlates with spindle-rich oscillations during sleep in rodents

Julie Seibt[1,2], Clément J. Richard[1], Johanna Sigl-Glöckner[3], Naoya Takahashi[4], David I. Kaplan[1], Guy Doron[4], Denis de Limoges[5], Christina Bocklisch[4] & Matthew E. Larkum [4]

How sleep influences brain plasticity is not known. In particular, why certain electro-encephalographic (EEG) rhythms are linked to memory consolidation is poorly understood. Calcium activity in dendrites is known to be necessary for structural plasticity changes, but this has never been carefully examined during sleep. Here, we report that calcium activity in populations of neocortical dendrites is increased and synchronised during oscillations in the spindle range in naturally sleeping rodents. Remarkably, the same relationship is not found in cell bodies of the same neurons and throughout the cortical column. Spindles during sleep have been suggested to be important for brain development and plasticity. Our results provide evidence for a physiological link of spindles in the cortex specific to dendrites, the main site of synaptic plasticity.

[1] NeuroCure Cluster of Excellence, Charité-Universitätsmedizin, D-10117 Berlin, Germany. [2] Surrey Sleep Research Centre, University of Surrey, GU2 7XP Guildford, UK. [3] Bernstein Center for Computational Neuroscience Berlin, Humboldt-Universitätzu Berlin, D-10115 Berlin, Germany. [4] Institute for Biology, Humboldt-Universität zu Berlin, D-10117 Berlin, Germany. [5] Department of Physiology, Universität Bern, 3012 Bern, Switzerland. Correspondence and requests for materials should be addressed to J.S. (email: j.seibt@surrey.ac.uk) or to M.E.L. (email: Matthew.larkum@gmail.com)

Accumulating evidence suggests a central role for sleep in brain plasticity consolidation, a process that enables the long-term storage of newly acquired information into brain networks[1–4]. Sleep is a complex brain state that alternates between periods of rapid eye movement (REM) and non-REM (NREM) sleep which are both characterised by specific electro-encephalographic (EEG) signatures. While REM and NREM sleep have both been implicated in the consolidation of various forms of brain plasticity and memories during development[5,6] and adulthood[1,7], the underlying mechanisms remain poorly understood.

Since dendrites receive the vast majority of synaptic inputs and have intrinsic functional properties themselves, they constitute the most likely physical substrate for brain plasticity and memory consolidation[8]. Recent studies using two-photon imaging and electron microscopy in rodents have revealed an important role for sleep in structural plasticity and show that dendritic spine formation and pruning induced by experience is facilitated by sleep and prevented by sleep deprivation during both develop-ment[9–11] and adulthood[11–13]. However, which aspects of sleep are involved remain unclear. Consolidation of structural plasticity in the motor cortex seems to involve increased dendritic calcium ($Ca^{2+}$) activity during REM sleep[11]. However, of the various EEG rhythms, slow-wave activity (SWA, 0.5–4 Hz) and spindles (9–16 Hz) during NREM sleep have been proposed to play a key role in synaptic remodelling associated with memory consolidation[1,14] but the underlying substrates and mechanisms have not yet been identified.

We therefore performed simultaneous EEG and calcium ($Ca^{2+}$) recordings from the dendrites of layer 5 (L5) cortical pyramidal neurons; the dendrites that are most closely associated with the generation of the EEG signal[15]. Using one-photon fibre-optic $Ca^{2+}$ imaging of dendritic populations[16], we show that increases in $Ca^{2+}$ activity correlate with oscillations in the spindle-rich sigma (9–16 Hz) and beta (16–30 Hz) frequency ranges. Inter-estingly, $Ca^{2+}$ activity was not associated with slower EEG oscillations (i.e., SWA). Two-photon imaging of single apical shaft dendrites confirms this result and further suggests that these oscillations reflect the synchronisation of dendritic activity. A similar relationship was not detected in the $Ca^{2+}$ activity of cell bodies in layers 2/3 (L2/3) and was significantly reduced in L5 neurons. Electrical recordings directly from the cell bodies of L5 pyramidal neurons further show that neuronal spiking activity was not affected by spindle events and correlated preferentially with oscillations in the delta band. These results suggest that pyramidal cell output is decoupled from dendritic activity during sleep spindles. Since spindles are known to be important for cognitive function, including memory formation, our results propose that dendritic $Ca^{2+}$ synchronisation serves a physiolo-gical mechanism underlying cortical plasticity during spindles in natural sleep.

## Results

**Combined $Ca^{2+}$ and EEG recordings in freely behaving rats.** To measure dendritic activity in freely behaving animals, we devel-oped a method for combined $Ca^{2+}$ imaging and EEG recordings in non-restrained rats (Fig. 1a). $Ca^{2+}$ changes in populations of dendrites were detected using a one-photon fibre-optic imaging approach[16–19] combined with the local injection of $Ca^{2+}$ indica-tors into L5 of the cortex[16] (Fig. 1b). All our recordings were performed during the light phase (between ZT6 and ZT12), when sleep dominates in rodents (Fig.1c). Using standard criteria from both fronto-frontal (FF) and fronto-parietal (FP) EEGs, we could identify five different behavioural states (Fig. 1d): active wake (AW), quiet wake (QW), non-rapid eye movement (NREM),

intermediate stage (IS) and REM sleep. IS is a short (44.43 ± 1.48 s, n = 28 rats) "transitional" sleep state found at the end of a NREM episode in cats and rodents, the main signature of which is an increase in spindle/sigma (9–16 Hz) and hippocampal theta (5–9 Hz; detected in the parietal EEG) activity[20,21] and a con-comitant decrease in delta/SWA oscillations (Supplementary Fig. 1b). The specificity of our dendritic recordings was controlled by measuring changes in $Ca^{2+}$ activity from the surrounding area (i.e., L2/3) and from animals that did not express any $Ca^{2+}$ indicator (Ctrl) to control for non-dendritic and background signals (e.g., autofluorescence, movement artefacts), respectively (Fig. 1e). Most of the recordings were performed using a non-invasive vertically oriented cannula for imaging placed directly above the cortical surface to prevent damage to the cortex and preserve network connections (Fig. 1a). However, to control for the potential contribution from L5 cell body activity to our dendritic recordings, we also imaged a subset of L5 injected animals with a 90° angled prism, ensuring that the excitation (and emission) light was confined to the upper layers of the cortex (Fig. 1f). We analysed on average 2 h of stable and artefact-free recordings in each group from a total of 28 rats. Importantly, there was no difference in age or behavioural parameters between experimental groups (Fig. 1g and Supplementary Table 1) that could account for potential differences in $Ca^{2+}$ activity. A detailed description of animals and recording parameters in each group is reported in Supplementary Table 1.

**Activity in population of dendrites increases during IS.** $Ca^{2+}$ activity recorded with this method displayed oscillatory activi-ty[17–19,22] (Fig. 2a, d) with maximum power for frequencies between 0.1 and 1 Hz (Fig. 2b and Supplementary Fig. 2a), which was significantly decreased after application of $Ca^{2+}$ channel blockers (Supplementary Fig. 2b). We therefore used the power density (PD) between 0.1 and 1 Hz to compare the changes in $Ca^{2+}$ signal between groups across all five behavioural states. $Ca^{2+}$ changes in dendrites were the largest during the IS (Fig. 2c, d). In contrast, $Ca^{2+}$ activity recorded from neurons in L2/3, sur-rounding L5 dendrites, displayed the largest increase during explorative behaviour in AW (Fig. 2c). Recordings from Ctrl animals did not reveal any changes in background signal across states (Fig. 2c). Finally, these results did not depend on the orientation of illumination or the $Ca^{2+}$ indicator used (Supple-mentary Fig. 3), supporting a biological and not methodological influence. As a complementary approach to our spectral analysis, we also performed a transient-based analysis of the $Ca^{2+}$ signal (see Methods). This method confirmed the absence of signal in the Ctrl group compared to dendritic and L2/3 $Ca^{2+}$ recordings which showed comparable number of detected transients across states (Supplementary Fig. 4a, b). Despite a similar trend of higher transient frequency during IS in both dendritic and L2/3 recordings (Supplementary Fig. 4c), we found a significantly larger proportion of transients with higher amplitude in dendrites during the IS (Supplementary Fig. 4d). These results suggest that the increase in both the amplitude and the frequency of $Ca^{2+}$ transients likely contributed to the increase in dendritic $Ca^{2+}$ PD during IS observed using the spectral analysis (Fig. 2c).

IS is a sleep state that is either ignored in most rodent studies or only studied when transitioning to REM sleep[23]. Our data confirm that the IS does not always transition to REM sleep as it occurs more frequently than REM sleep (Fig. 1f) as previously reported[24,25]. In fact, IS is often interrupted by brief microarousals quickly followed by another NREM sleep episode (see example in Fig. 2d) in the majority of the cases[21] (transitions from IS to (in %): NREM=45 ± 4.2, REM=31.3 ± 4.3, QW=19.7 ± 3.5, AW=4 ± 1.0; n = 28 rats). These successive

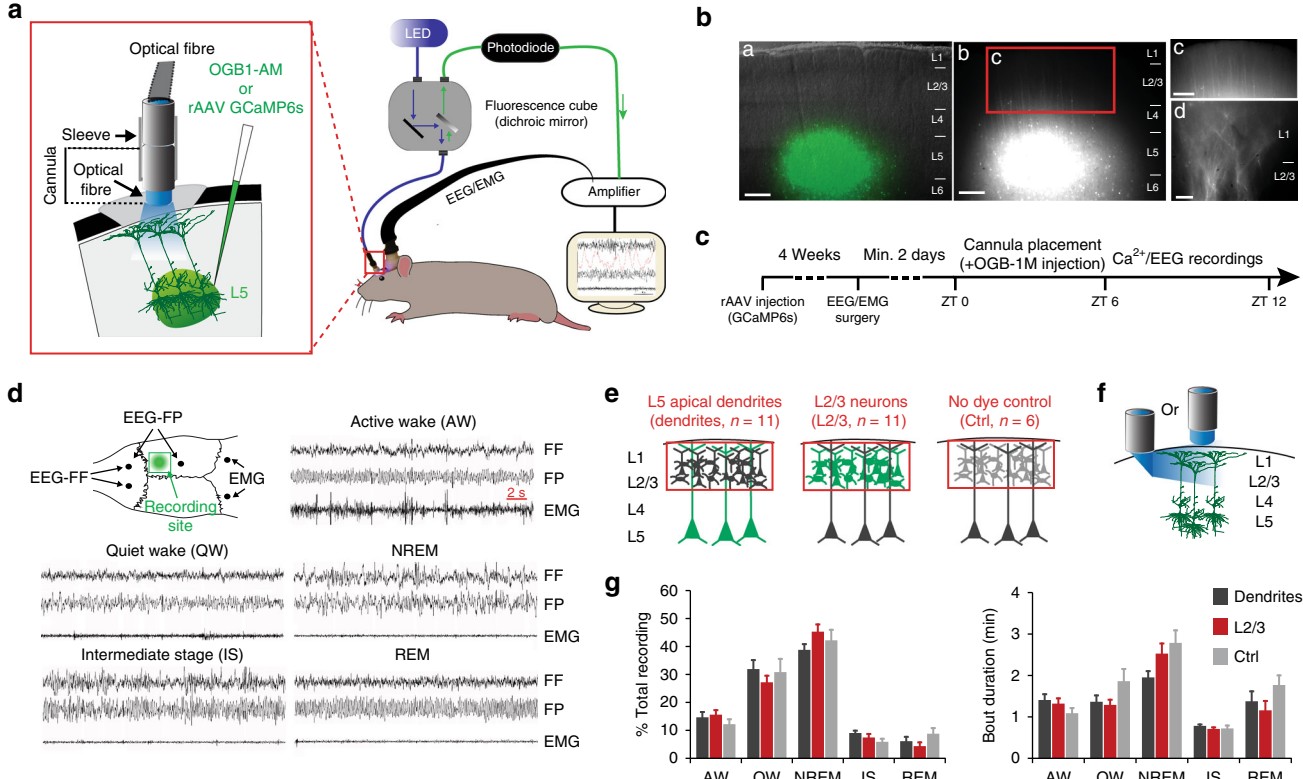

**Fig. 1** Ca$^{2+}$ imaging and EEG recordings in freely behaving rats. **a** Schematic of combined fibre-optic Ca$^{2+}$ imaging and EEG recordings. (*Left*) Enlarged view of the cannula attachment site. **b** Labelling of dendrites of L5 neurons with GCaMP6s. The injection site is shown on a 300 μm thick parasagittal slice from a rat at ~4 weeks after in vivo viral injection (a: superimposed with a brightfield image of the slice; b–d: dendrites expressing GCaMP6s at different magnifications). Layers are represented on the right. *Scale bars*, a–c=250 μm and d=20 μm. **c** Experimental design. **d** EEGs (fronto-frontal (FF), fronto-parietal:(FP)), EMG electrode placement and Ca$^{2+}$ recording site (*upper left*) and representative EEG/EMG traces for all five behavioural states. **e**, **f** Schematic representations of the region imaged (*red rectangle*) in the three main groups (**e**) and the two types of illumination orientations for dendritic recordings (**f**). **g** Mean (±s.e.m.) percentage of time spent in each behavioural state (*left*) and state bout duration (*right*) did not differ between groups (two-way ANOVA; factor "*group*": %: F$_{2, 150}$ = 0.26, P = 0.77; duration: F$_{2, 150}$ = 0.295, P = 0.055, see **e** for animal number/group)

NREM–IS periods most likely represent what has been recently defined as "NREM packets" within longer NREM episodes[21]. The idea that IS is an integral part of NREM sleep is further supported by our finding that IS parameters (i.e., distribution of oscillatory activity) and Ca$^{2+}$ activity were similar whether IS transitioned to wake, NREM, or REM sleep states (Supplementary Fig. 5). For these reasons, we combined NREM and IS into slow-wave-sleep (SWS) for further analysis. Recent work in the hippocampus and cortex suggests that neuronal activity can significantly change over the course of single wake and sleep episodes[21, 26]. Using a similar analysis as Grosmark et al.[26], we compared the dynamics of Ca$^{2+}$ activity in our three groups within individual WAKE (AW+QW), SWS (NREM+IS), and REM sleep episodes (Fig. 2e and see Methods). While WAKE episodes were associated with a general decrease in Ca$^{2+}$ activity in both dendrites and L2/3 neurons, Ca$^{2+}$ activity increased during sleep episodes only in dendrites, with the most pronounced increase during SWS episodes (Fig. 2f).

**Dendritic activity correlates with sigma–beta EEG power.** The finding that dendritic activity increases during spindle-rich sleep (IS) supports previous computational models that propose specific increases in dendritic Ca$^{2+}$ activity in neocortical pyramidal neurons during spindle activity[27]. However, this hypothesis has never been tested during natural sleep. We therefore focused first on the spindle-rich sigma frequency band (9–16 Hz). Using correlation of PD changes in individual 4 s epochs across

behavioural states, we found that sigma power fluctuations were strongly correlated with Ca$^{2+}$ activity in dendrites during SWS, but not with the activity in nearby L2/3 neurons (Fig. 3a, b), suggesting that this relation is dendrite specific. Surprisingly, we found an equally strong correlation between sigma PD and dendritic activity during REM sleep that was also significantly higher compared to activity in L2/3 (average correlation vs. sigma: r(dendrites)=0.44 ± 0.05, r(L2/3)=0.13 ± 0.04, P < 0.001, Holm-Sidak test, n = 11/group). Finally, we found that sigma activity during SWS was also a good predictor of dendritic Ca$^{2+}$ activity during REM sleep for a given animal, an effect that was again not found for L2/3 neurons (Fig. 3c). This result is consistent with recent data showing a role for both NREM and REM sleep for structural plasticity in spines[11, 12] and supports a functional relationship between SWS spindles and REM sleep as previously found in the context of hippocampal plasticity[3].

To test whether the relationship between dendritic Ca$^{2+}$ and sigma oscillations was specific, we correlated the change in Ca$^{2+}$ with the changes of a range of other frequency bands between 0.5 and 100 Hz during SWS (Fig. 4a). We found that beta (16–30 Hz) oscillations in both frontal and parietal EEGs were also highly correlated with dendritic activity during SWS, while no specific trend was observed for L2/3 recordings (Supplementary Fig. 6a). For a more precise and time-sensitive evaluation (than the 4 s epoch scoring system), we also compared the EEG and Ca$^{2+}$ signals using a time–frequency cross-correlation analysis adapted from Lachaux et al.[28] (Fig. 4b). Here, only the EEG surrounding

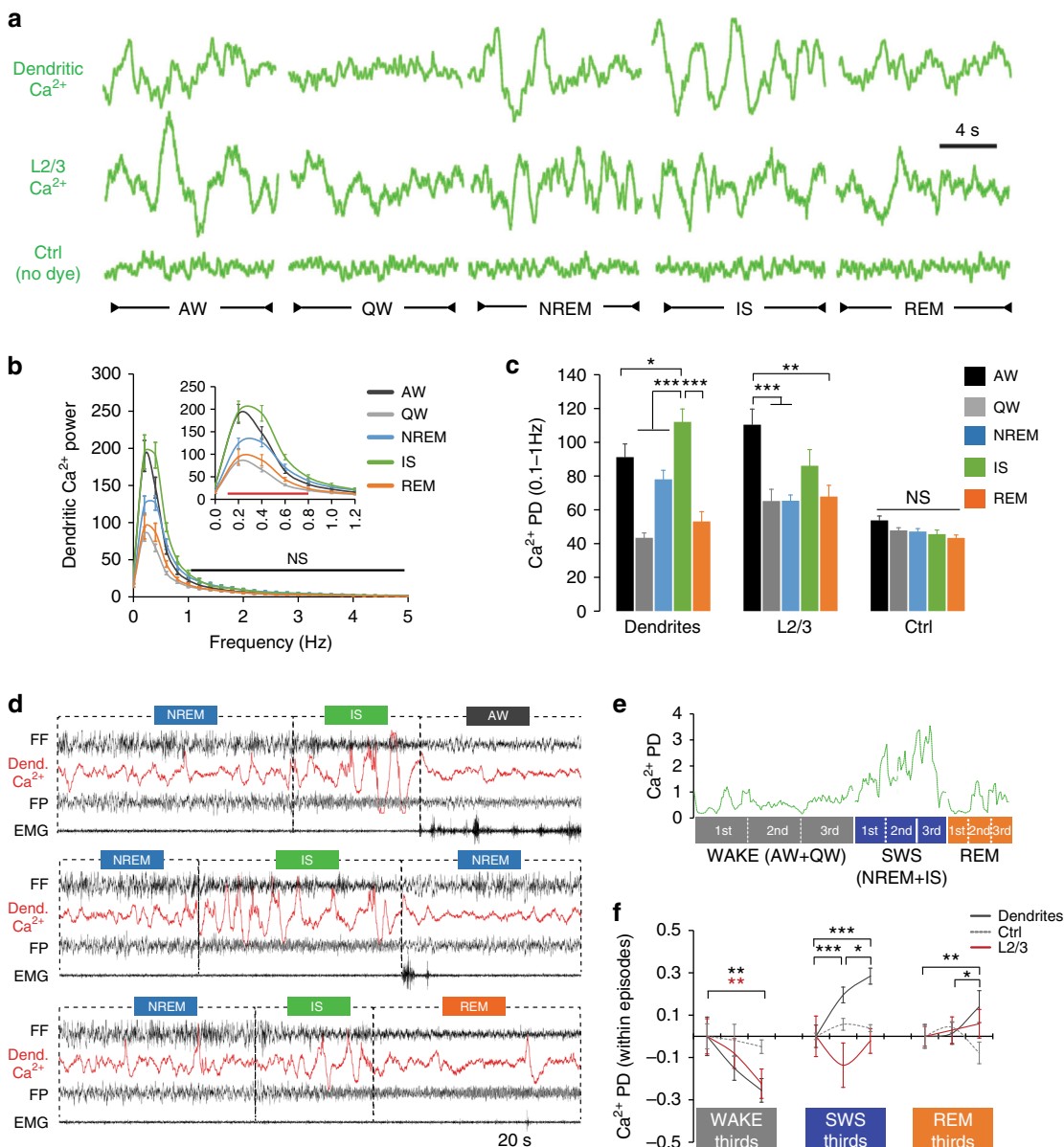

**Fig. 2** Ca²⁺ activity in populations of dendrites is largest at NREM transitions (IS). **a** Representative examples of fibre-optic signals recorded in the three groups for each behavioural state. **b** Mean (±s.e.m.) power spectra of the dendritic Ca²⁺ signal in each state (normalised to the mean across behavioural states). Significant differences between states were only found for frequencies <1 Hz (two-way ANOVA; factor "state": $F_{4, 1248} = 50.6$, $P < 0.001$; *red line* (inset graph): $P < 0.05$, *black line* (main graph): $P > 0.05$, Holm–Sidak test). **c** Mean (±s.e.m.) power density (PD) in the 0.1–1 Hz frequency band of the fibre-optic signal in the three groups across behavioural states (factor "state": $F_{4, 44(dendrites)} = 26.1$, $P < 0.001$ three-way ANOVA; $F_{4, 44(L2/3)} = 7.51$, $P < 0.001$ two-way ANOVA; $F_{4, 20(Ctrl)} = 2.67$, $P = 0.062$ two-way ANOVA; $*P < 0.05$, $**P < 0.01$ and $***P < 0.001$, Holm–Sidak test). The effects of the "dye" and "illumination" factors for dendritic and L2/3 Ca²⁺ activity are reported in Supplementary Fig. 3. **d** Examples of combined EEG/dendritic Ca²⁺ recordings at the IS transitions to AW, NREM and REM sleep. **e** Behavioural episodes analysis method (see Methods). An average PD is calculated for each third across individual episodes of WAKE (AW+QW), SWS (NREM+IS) and REM sleep. The *green line* represents fluctuations of dendritic Ca²⁺ PD. **f** Mean (±s.e.m.) optical signal PD changes within episode thirds. Statistical significance was tested using one-way RM ANOVA. WAKE: $F_{2, 20(dendrites)} = 6.57$, $F_{2, 20(L2/3)} = 6.8$, $P = 0.006$ for both groups; SWS: $F_{2, 20(dendrites)} = 23.76$, $P < 0.001$; REM: $F_{2, 16(dendrites)} = 7.3$, $P = 0.006$. $*P < 0.05$, $**P < 0.01$ and $***P < 0.001$, Holm–Sidak test. $N = 11$ for dendrites and L2/3 groups and $n = 6$ for the Ctrl group

the recording site (i.e., FP) confirmed the dendrite-specific correlation between sigma–beta oscillations and the Ca²⁺ signal (Fig. 4c and Supplementary Fig. 6b) and suggests a stronger link between local cortical networks and dendritic activity. This trend was further confirmed by examining the relationship between the magnitude of increase (ΔPD, Fig. 4d) in dendritic Ca²⁺ and EEGs oscillations during individual SWS episodes. Most oscillations

increase their power during SWS, with the largest increase seen for sigma as shown previously in the hippocampus and cortex (Supplementary Fig. 7)[21, 26]. We took advantage of the variability of dendritic activity dynamic between animals and correlated the average ΔPD of dendritic Ca²⁺ and EEG for each animal. Remarkably, changes in the sigma–beta frequency bands of the same individuals strongly correlated with these differences in

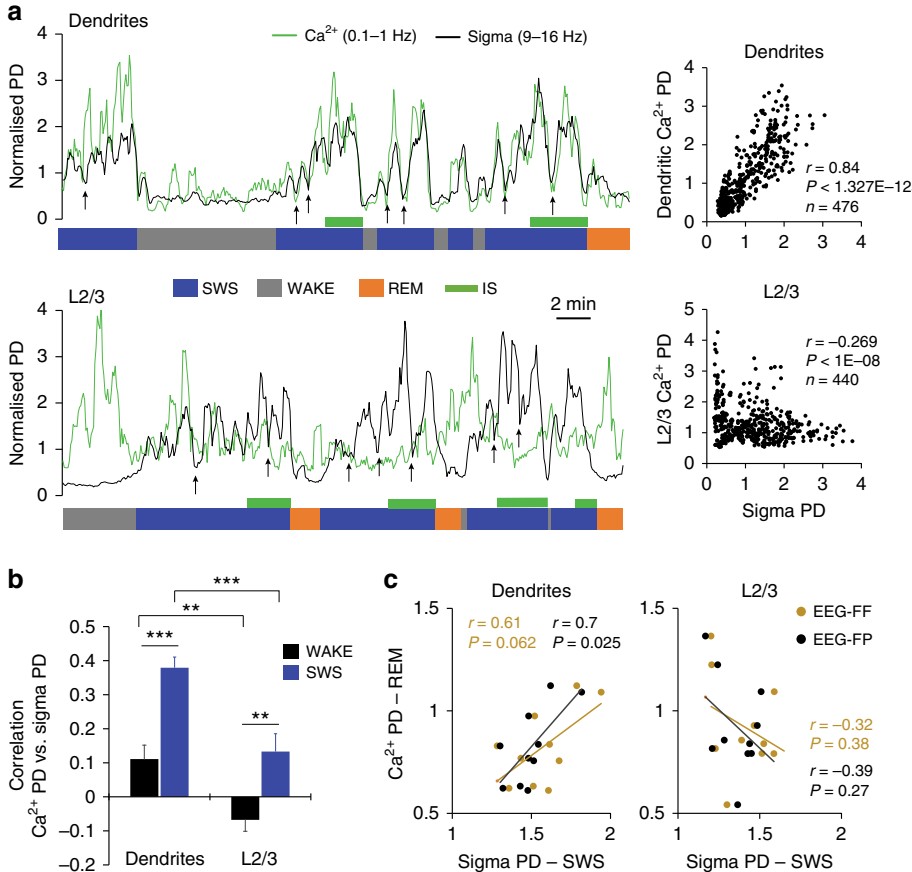

**Fig. 3** Correlation between $Ca^{2+}$ activity in population of dendrites and sigma EEG power. **a** Example of $Ca^{2+}$ and sigma PD fluctuations for an ~30 min recording segment from dendrites (*upper graph*) and L2/3 neurons (*lower graph*). Data are represented as trendlines (see Methods) and *arrows* point at microarousals (<5 epochs). Corresponding *scatter plots* are on the *right* (n=number of epochs, r=Pearson's correlation coefficient). **b** Mean (±s.e.m.) correlation between sigma and $Ca^{2+}$ activity in dendrites and L2/3 neurons across animals (n=11/group) for WAKE (AW and QW) and SWS. Correlation with FF and FP EEGs were pooled due to similar trend (two-way ANOVA; factor "EEG": $F_{1, 122} = 0.06$, $P = 0.801$; "Group": $F_{1, 122} = 33.26$, $P < 0.001$). Within groups, the correlation was higher during SWS compared to wakefulness (two-way ANOVA; factor "state": $F_{2, 120} = 22.97$, $P < 0.001$, **$P < 0.01$ and ***$P < 0.001$, Holm–Sidak test). **c** Correlation (Pearson) between sigma PD during SWS and $Ca^{2+}$ PD in dendrites and L2/3 neurons during REM sleep. Values represent the average $Ca^{2+}$ and EEG (FF and FP) PDs for each animal. Values from individual 4 s epochs were used for all correlations

$Ca^{2+}$. This relationship was only significant for local EEG (i.e., FP, Fig. 4e, f) and absent for L2/3 and Ctrl recordings (Supplementary Fig. 8). Taken together, these results not only suggest that the relationship between sigma–beta oscillations is specific to dendrites, but they also imply that local EEG in this frequency band can be used as a biomarker of large-scale dendritic activity across specific cortical areas.

**$Ca^{2+}$ activity in single dendrites and somata of L5 neurons.** Changes in $Ca^{2+}$ detected using our fibre-optic imaging method capture the summation of hundreds of dendrites and are therefore well suited to comparison with the EEG signals. But how do these changes in $Ca^{2+}$ manifest at the single-cell and single-dendrite level? To investigate this we first combined two-photon $Ca^{2+}$ imaging of single apical shaft dendrites and somata of L5 neurons with EEG recordings in mice habituated to sleep in a head-fixed apparatus (Fig. 5a and Supplementary Fig. 9a, b). We used mice instead of rats as chronic two-photon imaging is not possible in rats due to the presence of a thicker dura. The same labelling method (i.e., injection of GCaMP6s to L5) was used for these experiments. We imaged 142 apical shaft dendrites and 89 somata (n = 3 mice/group, see Methods for description of imaging depths in each group). Our imaging sessions lasted

on average 1.5 h in each group (dendrites: 101.5 ± 9.8 min; L5 somata: 99.6 ± 21.2 min). The distribution of behavioural states was comparable to our freely behaving experiments despite a higher amount of QW (Supplementary Fig. 9c). As in the fibre-optic recordings, two-photon imaging showed that $Ca^{2+}$ activity in dendrites of L5 neurons was highest during IS and lowest during wakefulness (Fig. 5b). In contrast with L2/3 neurons, $Ca^{2+}$ in L5 cell bodies also increased during sleep showing maximum activity during REM sleep (Fig. 5b).

However, correlation between sigma PD and $Ca^{2+}$ activity in single dendrites and somata was quite variable showing both positive and negative correlations (Fig. 5c–e). We therefore partitioned single dendrite and soma into two categories: with positive ($r > 0$) or negative ($r < 0$) correlations with sigma PD. The majority of dendrites and somata showed a positive correlation with sigma power (dendrites=75.35%; somata=77.53%, Fig. 5f). This partitioning revealed the same relationship between dendritic $Ca^{2+}$ activity and different frequency bands during SWS as the population recordings for $r > 0$ and the inverse relationship for $r < 0$ (Fig. 5e, f). For the somata, a similar trend was observed only for positive correlations and was significantly smaller than for dendrites (Fig. 5f). However, this trend was not found when $Ca^{2+}$ imaging was

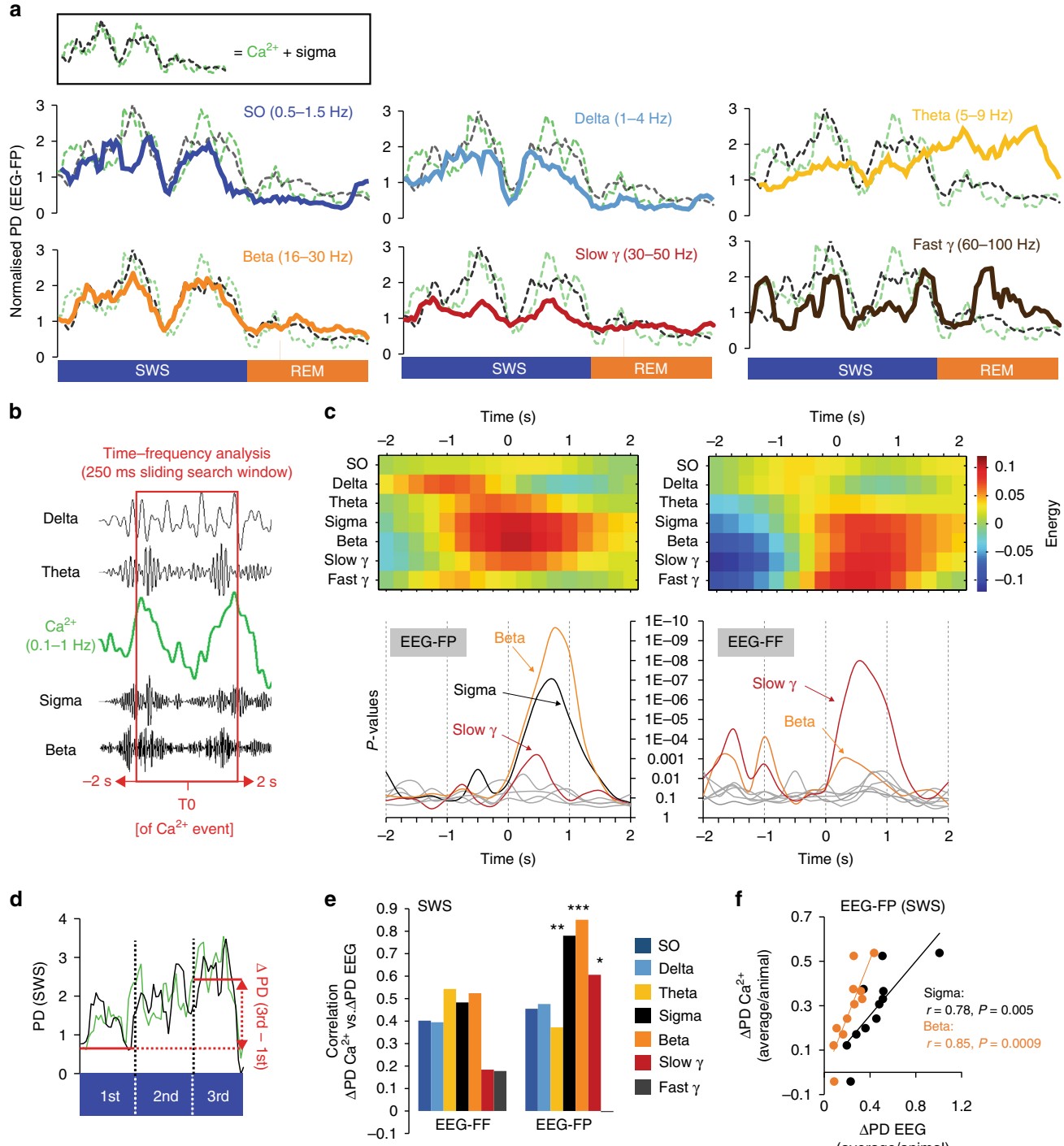

**Fig. 4** Changes in Ca²⁺ activity in population of dendrites correlates with local changes in sigma–beta power during SWS. **a** Examples of PD time course for different frequency bands from the EEG surrounding the Ca²⁺ imaging site (i.e., EEG-FP). Traces are plotted against the sigma+dendritic Ca²⁺ PD trendlines (*dotted lines, inset*). **b** Illustration of the cross-correlation time–frequency analysis (see Methods). **c** Time–frequency analysis for dendritic recordings. Results for local (FP, *left*) and distant (FF, *right*) EEGs are represented. Energy heat maps (*upper graphs*) with corresponding *P*-values (*bottom graphs*) for a ±2 s cross-correlation time window (see Supplementary Fig. 6b for L2/3 results). **d** Illustration of the ΔPD (PD 3rd−PD 1st) calculation. **e** Correlation coefficient between the mean (across SWS episodes in each animal) Ca²⁺ and EEG frequency bands ΔPDs (n = 11). Results are shown separately for frontal and parietal EEGs. *P < 0.05, **P < 0.01 and ***P < 0.001, Pearson's correlation. **f** Corresponding scatter plots for sigma and beta frequency bands for EEG-FP (values represent individual animals). Correlations were not significant for frontal EEG (vs. EEG-FF, sigma: r = 0.48, P = 0.13; beta: r = 0.52, P = 0.1)

restricted to L5 pyramidal neurons (Supplementary Fig. 10). This suggests that our L5 somatic data might be contaminated by responses from nearby inhibitory neurons which have been shown to increase their activity during spindles in the cortex[31].

**Sigma–beta power reflects dendritic activity synchronisation.** Since changes in EEG are thought to reflect coherent synaptic inputs[15], we were particularly interested in the synchrony of the Ca²⁺ changes and how those changes related to the EEGs. While we found that changes in dendritic Ca²⁺ activity were

significantly more asynchronous across behavioural states than activity of cell bodies (Fig. 6a), activity synchronisation in both compartments of L5 neurons increased during sleep compared to waking states with highest values found during IS (Fig. 6b, c). Remarkably, $Ca^{2+}$ activity synchronisation in populations of dendrites was specifically correlated with sigma–beta PD changes during SWS, while no specific correlation trend was observed for somatic activity synchronisation (Fig. 6d, e). The data at the single soma and dendrite level therefore confirm and extend the results obtained at the population level. They further suggest that sigma–beta oscillations reflect synchronised $Ca^{2+}$ activity in L5 neurons specifically in dendrites.

**Spiking of L5 cell bodies is not influenced by spindles.** Since increases in dendritic $Ca^{2+}$ activity in L5 neurons is often linked to increased firing at the cell body[29], we recorded somatic spiking

activity by performing juxtacellular recordings from L5 somata combined with EEG/LFP recordings in head-fixed rats (Fig. 7a). We recorded a total of 23 L5 somata for which we could identify clear spindle events in the local field potential (LFP) (Fig. 7b). On average, firing rate remained quite stable across behavioural states and no significant changes were detected during spindles (one-way RM ANOVA, F = 0.842, P = 0.48, Fig. 7c). Since spindles are short events (between 0.5 and 3 s) which are difficult to compare with longer wake and sleep states (i.e., WAKE, SWS and REM sleep), we also compared firing rates during spindles with the 2 s immediately before ("pre") or after ("post") spindle events for each cell (Fig. 7d). This analysis revealed a nonsignificant trend towards higher firing rate during the 2 s preceding a spindle event (one-way RM ANOVA, $\chi^2 = 2.88$, P = 0.24). Finally, a cross-correlation analysis between EEG/LFP and firing rate during SWS revealed that, compared to other frequencies, delta oscillations were the best predictor of L5 somatic firing (Fig. 7e

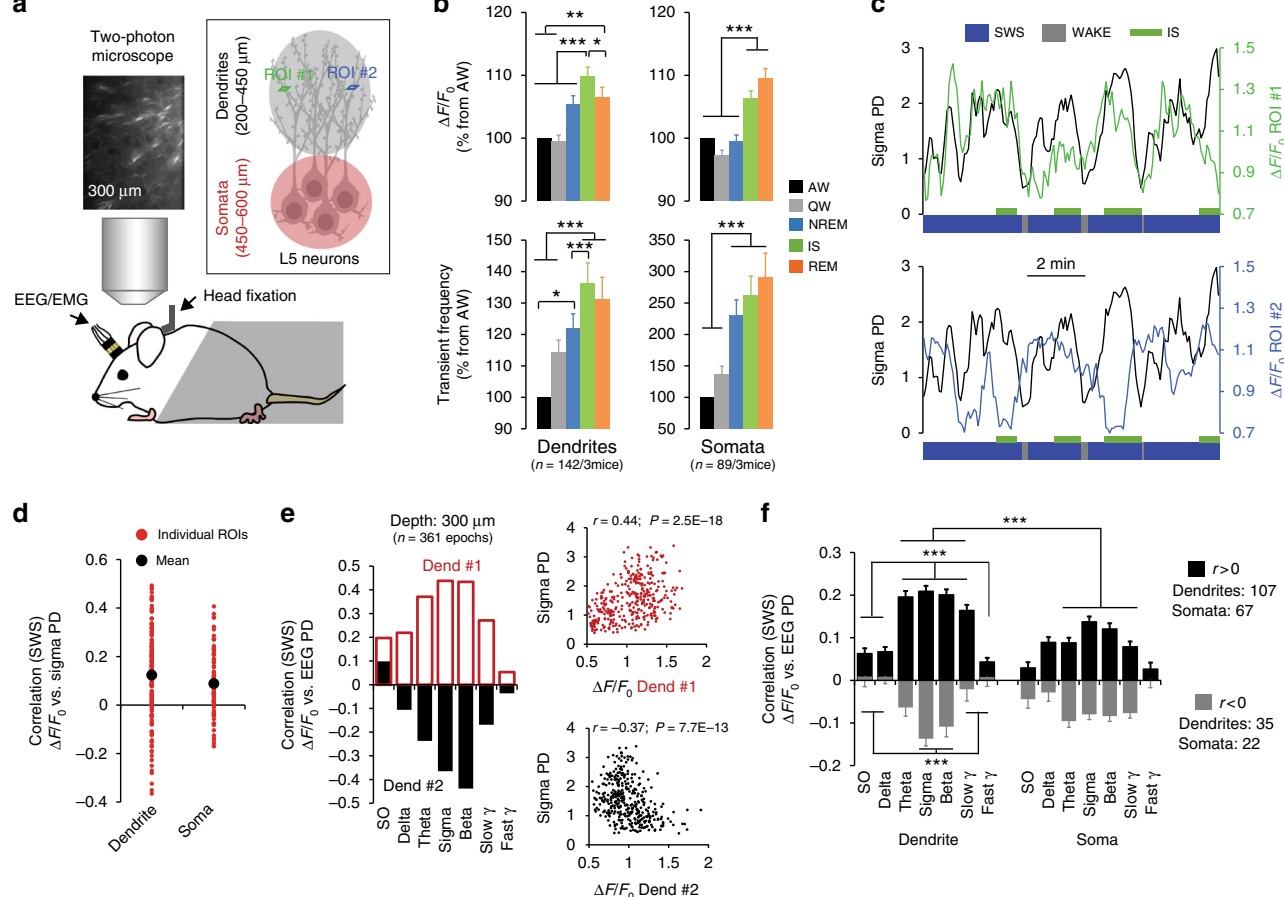

**Fig. 5** Spindle–beta oscillations correlate with increased and decreased $Ca^{2+}$ activity in single dendrites. **a** Schematic of combined EEG recordings and two-photon $Ca^{2+}$ imaging of apical dendrites and somata of L5 neurons. **b** Mean (±s.e.m.) $Ca^{2+}$ activity ($\Delta F/F_0$: upper graphs; transient frequency: lower graphs) across single dendrites (left graphs, n = 143 dendrites in 3 mice) and single somata (right graphs, n = 89 somata in 3 mice) for each behavioural state. Activity in individual dendrites and somata are expressed as percentage of change from respective activity during AW (one-way RM ANOVA; Dendrites: $F_{4,722} = 25.32$; Somata: $F_{4,307} = 22.79$; P < 0.001 for both groups, *P < 0.05, **P < 0.001 and ***P < 0.01, Holm–Sidak test). **c** Examples of $Ca^{2+}$ $\Delta F/F_0$ fluctuations in regions of interest (ROIs) from two dendrites imaged simultaneously (depth: −200 μm, schematised in (**a**)) and sigma PD. Note the reverse correlation trend. **d** Correlation (Pearson) between $\Delta F/F_0$ for all single dendrites and somata and sigma PD during SWS. **e** Example of correlation (Pearson) coefficient between $\Delta F/F_0$ and frequency bands (EEG-FP) for two individual dendrites recorded simultaneously at −300 μm below the pia (n = number of SWS epochs used for correlation). Corresponding scatter plots for sigma PD is shown on the right. **f** Mean (±s.e.m.) correlation across dendrites and somata selected to have a positive (r > 0) or negative (r < 0) correlation with sigma PD. Dendrites show a stronger correlation with EEG changes compared to somata, especially for positive correlations (two-way ANOVA; factor "group": r > 0: $F_{1,1218} = 59.71$, P < 0.001; r < 0: $F_{1,371} = 1.56$, P = 0.21; ***P < 0.001, Holm–Sidak test). Within groups, increase and decrease of $Ca^{2+}$ activity in single dendrites correlated with frequency bands surrounding the sigma band (two-way ANOVA; factor "frequency": r > 0: $F_{6,1218} = 36.93$, P < 0.001; r < 0: $F_{6,371} = 8.16$, P < 0.001; ***P < 0.001, Holm–Sidak test). Values from individual 4 s SWS epochs were used for all correlations

and Supplementary Fig. 11) and further confirm the absence of relationship between pyramidal cell activity and spindles in the cortex as previously shown[27, 31, 32]. We took advantage of our combined EEG/LFP recordings to also investigate the relationship between spindle events and underlying oscillatory activity. Our results demonstrate that increased power in both sigma and beta bands reflects increased spindle density. This relationship was similar for local (LFP) and more global (EEG) network activity measures (Fig. 7f) and suggests that spindles in the cortex, at least in rats, are represented by oscillations in a broader frequency band than the sigma band as previously thought. This result has also important implications for the interpretation of our data as it implies that the correlations between dendritic activity and sigma–beta oscillations in our one- and two-photon recordings are linked to underlying changes in cortical spindles.

## Discussion

Despite the central importance of local changes in dendritic $Ca^{2+}$ for plasticity and memory[8], this aspect had never been carefully examined during sleep (but see ref. [11]). Here, we measured $Ca^{2+}$

changes in population and single apical shaft dendrites of L5 pyramidal neurons in sleeping rodents using two independent methods (i.e., one- and two-photon imaging) as well as juxtacellular recordings from L5 pyramidal cell bodies, combined with EEG/LFP recordings. We found that $Ca^{2+}$ activity at the population and single dendrite levels not only varies across behavioural states but is increased and synchronised during spindle-rich sleep episodes, in particular the so-called IS of SWS. More specifically, we show that $Ca^{2+}$ activity synchronisation in dendrites is correlated with specific EEG oscillations in the sigma–beta frequency range during SWS. This specific correlation was observed across two different rodent species using two different imaging approaches. Our combined EEG and LFP recordings further revealed that sigma–beta oscillations were specifically linked to cortical spindle density in rats (Fig. 7f), which taken together with the dendritic $Ca^{2+}$ recordings implies that spindle events are linked to synchronisation of dendritic activity. Finally, we provide compelling evidence that this relationship is specific to dendrites as activity in L5 and L2/3 cell bodies did not reveal such correlations.

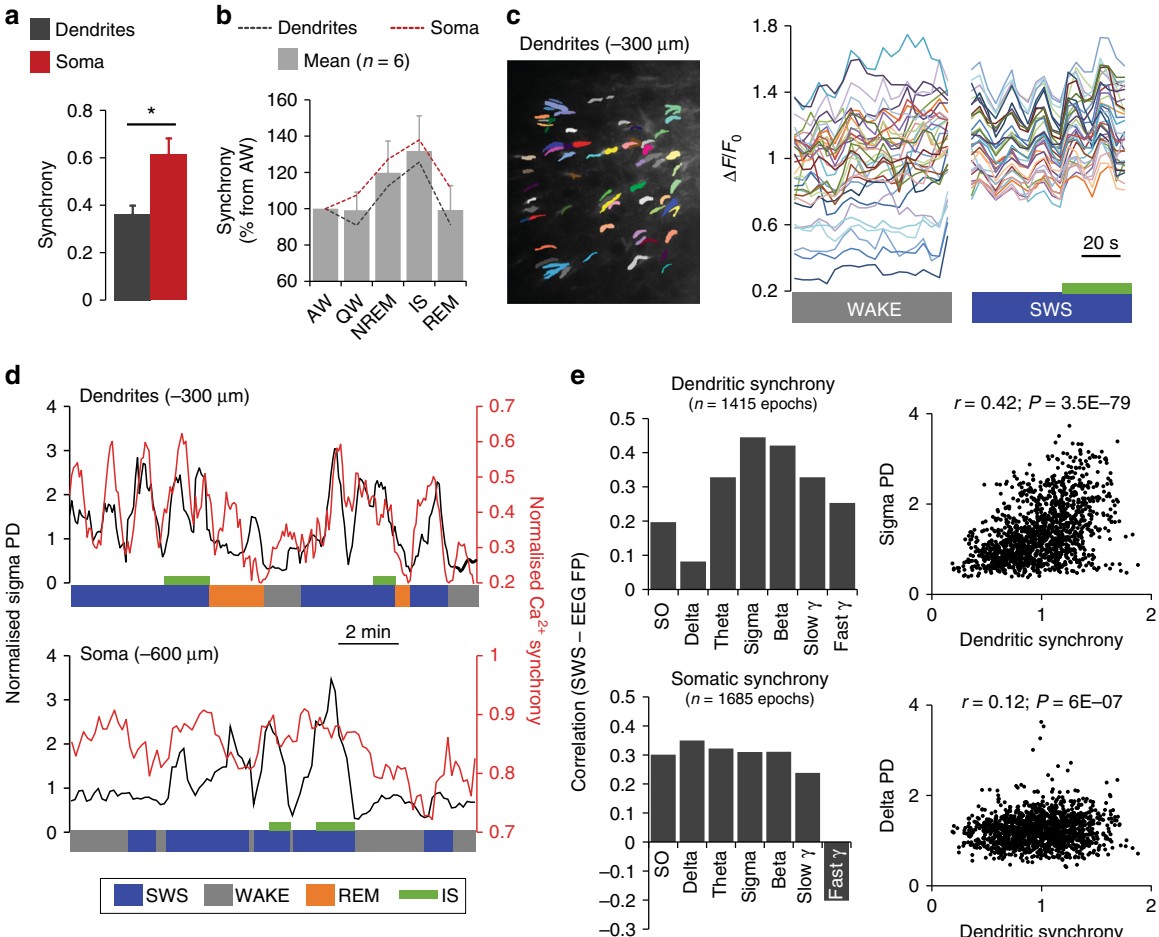

**Fig. 6** Spindle–beta oscillations reflect increased dendritic $Ca^{2+}$ activity synchronisation. **a** Average $Ca^{2+}$ activity synchronisation in dendrites and somata across all behavioural states ($H = 7.1$, $P = 0.008$ Kruskal–Wallis one-way ANOVA on ranks; *$P < 0.05$, Dunn's test). $N = 15$/group (5 states/mice, 3 mice/group). **b** Synchronisation of $Ca^{2+}$ activity in dendrites and somata of L5 neurons across behavioural states (expressed as percentage of respective AW value for each animal, $n = 11$/group). For **a**, **b**, values represent means ± s.e.m. **c** Example of synchronisation of $Ca^{2+}$ activity ($\Delta F/F_0$) of the same 34 dendrites (imaged at −300 μm) during SWS compared to WAKE. IS is represented by the *green bar*. **d** Examples of changes in sigma PD and $Ca^{2+}$ activity synchronisation for recordings from dendrites of L5 neurons (*upper graph*) and L5 somata (*lower graph*). **e** Correlation (Pearson) coefficient between dendritic (*upper graph*) and somatic (*lower graph*) synchrony of $Ca^{2+}$ activity and different frequency bands (EEG-FP). $N$ = number of SWS epochs used for correlations (values from the three animals in each group were pooled). Corresponding scatter plot for dendritic synchrony and sigma and delta band are represented. For **c–e**, values from individual 4 s SWS epochs were used for correlations

While spindles have been linked to cognitive functions, including memory, in humans and animals[14], the cellular mechanisms and function of cortical spindles remain largely unknown. Until now, electrophysiological recordings in humans and animals failed to show significant changes in cortical activity linked to spindles[27, 30, 31]. Our study is the first to reveal such a link, which is specific to cortical dendrites and may explain the absence of correlative data in previous studies. Our observation that this relationship reflects synchronisation of activity and is specific to dendrites has two important consequences. First, it supports the idea that activity synchronisation is an important component that shapes EEG signal, as recently shown in non-

human primates[32]. Second, it suggests that synchronised $Ca^{2+}$ changes in dendrites during spindles are decoupled from somatic output firing. The mechanisms and function of this decoupling are not clear. A specific link between spindles and dendritic $Ca^{2+}$ activity in L5 pyramidal neurons was previously proposed on the basis of recordings from anaesthetised animals and computational models[27, 33]. It has been hypothesised that the firing of deep-layer pyramidal neurons is suppressed during spindle activity because of the strong recruitment of inhibition[27]. While our study did not investigate the role of inhibition, our results combined with results showing increased inhibitory tone during spindles in rodents[31] lend experimental support for these

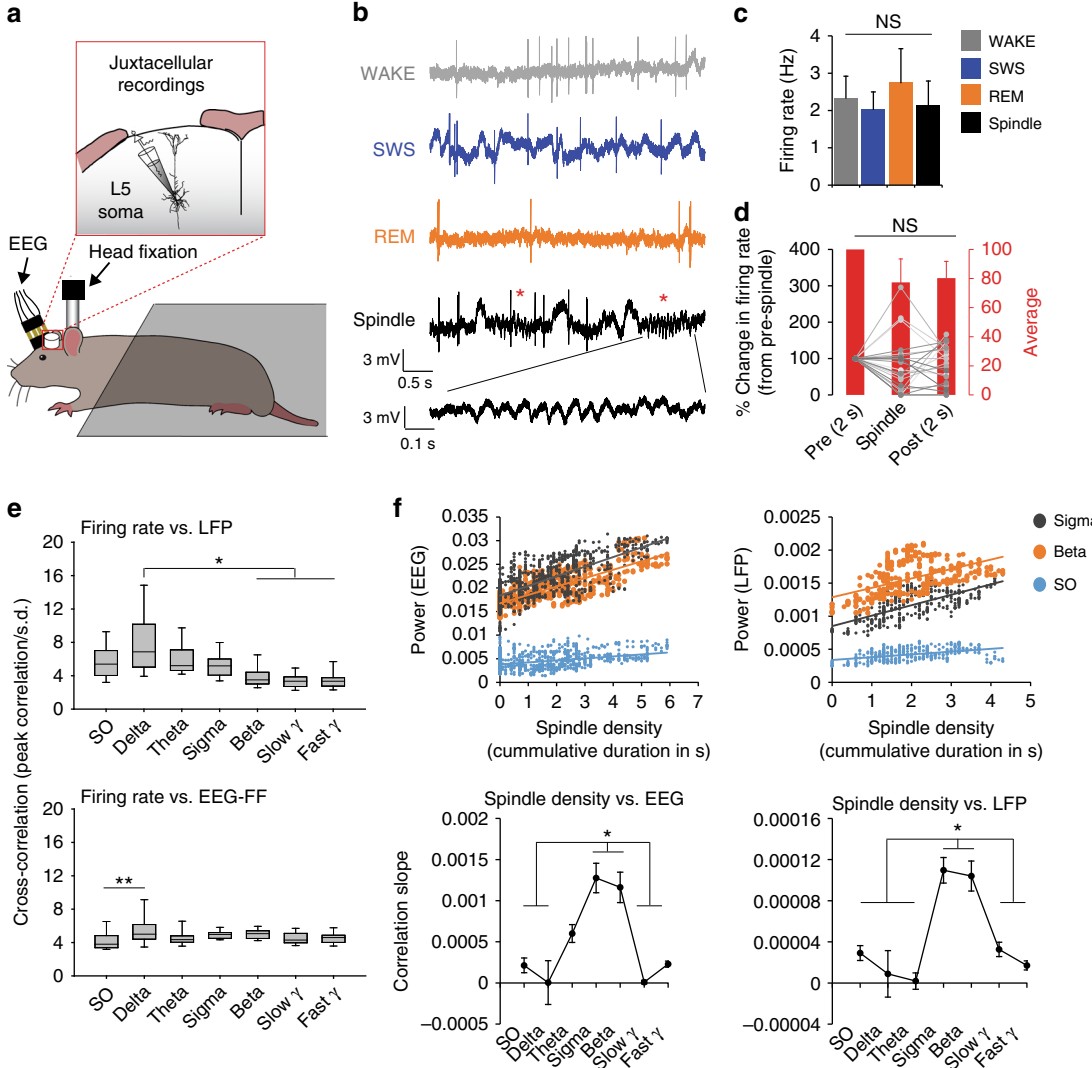

**Fig. 7** Relationship between EEG, LFP and spiking activity of L5 cell bodies. **a** Schematic of combined juxtacellular and EEG recordings in head-fixed rats. **b** Representative examples of local field potential (LFP) traces and firing pattern of L5 pyramidal neurons during WAKE, SWS, REM sleep and isolated spindles during SWS (*red asterisks*). **c** Mean ($\pm$ s.e.m.) firing rate across behavioural states and spindle events ($n = 23$ cells/2 rats). **d** Peri-spindle change in firing rate (expressed as percentage of change from pre-spindle events) for each cell. *Lines* represent individual cells and *red bars* the average across all cells. **e** Average cross-correlation between firing rate and LFP/EEG frequency bands during SWS (expressed as the peak correlation within a $\pm 5$ s window/s.d. of the cross-correlation; see Methods). Firing rate shows a stronger correlation with delta oscillations in the LFP (*upper graph*; $H = 55.2$, $P < 0.001$ Kruskal–Wallis one-way ANOVA on ranks; *$P < 0.05$, Dunn's test). A two-way ANOVA revealed a significant effect of EEG and frequency on firing rate (factor "EEG": $F_{1, 224} = 12.49$, factor "frequency": $F_{1, 224} = 4.69$, $P < 0.001$ for both factors). Post hoc comparisons showed a stronger correlation between firing and delta oscillations only in the frontal EEG (*lower graph*; **$P < 0.01$, Holm–Sidak test; see Supplementary Fig. 11 for parietal EEG results). **f** Sigma–beta oscillations reflect spindle density. (*Upper graphs*) Representative scatter plots showing correlation between spindle density (cumulative spindle duration within 10 s windows, see Methods) and EEG/LFP power for sigma, beta and SO for one recording (data points represent individual 10 s windows). (*Lower graphs*) Mean ($\pm$s.e.m.) correlation slope for all frequency bands for EEG and LFP recordings (EEG: $H = 49.9$, LFP: $H = 51.8$, $P < 0.001$ for both, Kruskal–Wallis one-way ANOVA on ranks; *$P < 0.05$, Dunn's test; $n = 17$, see Methods for number justification)

proposed mechanisms during natural sleep. Perisomatic inhibition of L5 pyramidal neurons during spindles also explains the different trends we observe in $Ca^{2+}$ (Fig. 5) and spiking (Fig. 7) activity in L5 cell bodies. Some L5 inhibitory neurons might have been labelled with GCaMP6s (driven by the synapsin promoter) and thus contributed to the increased correlation of $Ca^{2+}$ activity with spindle-rich oscillations during SWS. Our $Ca^{2+}$ imaging data of cell bodies in the Rbp4 (i.e., specific to L5 pyramidal cell) mouse line further support this interpretation (Supplementary Fig. 10). Future experiments using $Ca^{2+}$ imaging in transgenic lines combined with electrophysiology will be necessary to properly address this question as a decoupling of somatic and dendritic activity during spindles might reshape our views on dendritic electrogenesis as well as synaptic plasticity mechanisms.

In the context of plasticity, a decoupling of dendritic and somatic firing implies that spindle-related plasticity mechanisms for memory consolidation uses a local dendritic, non-Hebbian mechanism[34]. The influence of sleep and experience on dendritic functions ($Ca^{2+}$ activity and spine structure) has been the topic of several studies in recent years that suggest that sleep has an important influence on dendritic plasticity during development and adulthood[9–13]. However, how sleep stages participate in this function is less clear. A role for dendritic activity during SWS in memory consolidation is supported by a recent study that showed that inhibition of top-down inputs (mainly influencing dendrites[35, 36]) during SWS impairs perceptual learning[37]. Although a specific role for spindles (or other sleep oscillations) was not investigated in this study, their data support the idea that dendritic activation during SWS plays an important role for memory consolidation.

In a very recent study, Li et al.[11] showed that new spine stabilisation and pruning is favoured by REM sleep and is accompanied by increased $Ca^{2+}$ activity in apical tuft dendrites during that state. We found similar changes in $Ca^{2+}$ activity in our study using the same approach (i.e., two-photon imaging of single dendrites). Interestingly, when measured at the population level (i.e., using the fibre-optic approach) we saw a reduction in $Ca^{2+}$ fluctuations in REM vs. SWS (Fig. 2c). We also saw a similar reduction in synchronised $Ca^{2+}$ activity between these states (Fig. 6b). We hypothesise that the difference in $Ca^{2+}$ activity is explained by the sensitivity to synchronisation in the network of the two different (i.e., population vs. single) imaging approaches. Here, during REM sleep the general level of activity is high but asynchronous so that population recording approaches, such as EEG or fibre-optic imaging, result in steady signals analogous to a crowded room in which the ambient noise level appears not to fluctuate. On the other hand, when the fluctuations are synchronous (i.e., SWS), they are clearly detectable at the population level. In this respect, the fibre-optic data highlight the congruence between dendritic $Ca^{2+}$ and spindles in the EEG signal. Generally, the data we present highlight the importance of spindle-like oscillations in reflecting synchronisation of $Ca^{2+}$ activity in dendrites of L5 neurons that, in the context of memory, may prime specific dendrites for plasticity consolidation across the NREM–REM sleep cycle[1, 3].

Our data also raise the intriguing possibility of a functional continuum of a wider frequency band in the sigma–beta range (i.e., 9–30 Hz) related to cortical spindles during development and adulthood. So far, this frequency band was mostly associated with immature spindle oscillations during early development ("delta brush" (8–25 Hz) in humans and "spindle-burst" (5–25 Hz) in rodents[38]). However, our data show a specific correlation between sigma–beta oscillations with cortical spindle events in rodents which support previous data in adults that demonstrated a coupling of spindles with beta frequency in the neocortex of humans[39] and, more recently, in rodents[21]. This could lead to a reevaluation of the nature and origin of spindles in the cortex. While it is not clear if developmental and adult spindles represent similar phenomena, spindle-like oscillations during development are also known to be important for brain plasticity implicated in the (re)organisation of immature circuits[40]. Our results therefore provide a physiological substrate underlying the functional coupling between spindle and beta oscillations that may reflect, at the EEG level, the maturation and plasticity of dendrites across developmental stages.

In conclusion, our results suggest that EEG sigma–beta fluctuations can be used as a specific hallmark of cortical dendritic activity. The relationship between experience, spindle-beta/$Ca^{2+}$ coupling in dendrites and cognitive functions remains to be determined. Here, while the two-photon approach offers the best resolution, it is also more expensive and cumbersome to integrate with memory-related behavioural paradigms, whereas the fibre-optic approach offers a convenient and effective way to examine this question while still tracking state-dependent dendritic activity. Fundamentally, the correlation between sleep spindles and dendritic $Ca^{2+}$ demonstrated here suggests that the dendrites of L5 pyramidal neurons might be the locus of important mechanisms related to memory consolidation.

## Methods

**Animals**. All experiments and procedures were approved by the veterinary office of the canton of Bern, Switzerland, and the veterinary office of Landesamtfür Gesundheit und Soziales (LaGeSo) regulation in Berlin, Germany. We used female Wistar rats (P28–P52, Charles River) for freely behaving experiments and male Wistar rats for juxtacellular recordings. For two-photon experiments, we used female (>P40) C57BL/6 (Charles River) or Rbp4-cre (031125-UCD, MMRRC) mice. All animals were group-housed on a 12:12 light/dark cycle with ad lib food and water.

**In vivo loading of $Ca^{2+}$-sensitive dyes**. Activity from L2/3 and dendrites were monitored using the synthetic $Ca^{2+}$ dye Oregon Green 488 BAPTA-1 (OGB-1)-AM (Molecular Probes, Eugene, OR, USA) prepared as described in ref. [16] or the genetically encoded calcium indicator (GECI) GCaMP6s (AAV1.Syn.GCaMP6s. WPRE.SV40, PENN Vector Core). In the control group, half of the rats did not receive any injection and the other half received an injection of a control virus (AAV1.Syn.Flex.GCaMP6s.WPRE.SV40, PENN Vector Core) to mimic the injection procedure and potential follow-up effects (Supplementary Table 1). All injections were performed in the primary somatosensory cortex, centred on the hindlimb area as described in ref. [16]. Injections of GCaMP6s were performed in rat pups (P11-P14, injection depths: L5=1.1 mm; L2/3 and Ctrl=200 μm; 1 mm posterior from bregma and 1 mm from midline) to allow diffusion and expression of the virus into dendrites (3–4 weeks). Animals that were imaged using OGB1-AM were injected on the day of recording (Fig. 1c) using the same procedure with slightly different coordinates and injection depths adjusted for age (L5=1.5 mm; L2/3=250 μm, 1.5 mm posterior to bregma and 2.2 mm from midline[16]). Rat were anaesthetised with isoflurane (1.5–3%) and place in a stereotaxic frame. Body temperature was maintained at ~37 °C using a heating pad. A small incision was made in the skin and a hole was drilled through the skull above the somatosensory cortex. Between 30 and 50 nl of dye was pressure injected over 1 min, followed by a waiting period of 5 min before the micropipette (5 μl calibrated micropipettes, Blaubrand®) was then slowly removed. After virus injection, the site was covered with silicone (Kwik-Cast™, World Precision Instruments, Inc.) and the skin was sutured. At the end of the surgical procedure, buprenorphine was administered as a long-lasting analgesic (0.01 to 0.05 mg/kg, intraperitoneal (IP)) and the pups were returned to the mother.

**Surgeries for freely behaving recordings**. At least 2 days before the recording session, rats underwent surgery for EEG/EMG implantation under isoflurane anaesthesia (1.5–3% in $O_2$). Rat were placed in a stereotaxic frame and controlled for body temperature. Headmounts (Pinnacle Technology, Inc.) were used for FF and FP EEG recordings (Fig. 1d). After skin, blood and tissue covering the skull were removed, the bone was covered with light-curing adhesive (OptiBond, Kerr, Orange, CA, USA). No adhesive was applied to parts of the skull that were later drilled through for placement of the EEG wires. Three silver wires were used as EEGs and two stainless steel wires were implanted in the nuchal muscles for EMG recordings (Pinnacle Technology, Inc., USA). Electrodes were affixed to the skull using bone screws and dental acrylic. The area of the skull above the imaged cortical region was left exposed and covered with a protective thin layer of dental cement for cannula implantation on the day of recording (Fig. 1c). At the end of

the surgical procedure, buprenorphine was administered as a long-lasting analgesic (0.01 to 0.05 mg/kg IP) and the animals were allowed to recover for at least 2 days.

On the day of the experiment the animal was anaesthetised with isoflurane (1.5–3% in $O_2$) and placed in a stereotaxic frame for fibre-optic cannula placement. Virus-injected animals only underwent surgery for cannula implantation on that day. A small craniotomy was made (~1 mm$^2$). After a careful incision of the dura was made to expose a small area of the cortical surface (<0.5 mm$^2$), a subset of animals received an OGB1-AM injection (Fig. 1c). In all animals, a fibre-optic cannula was placed directly on the cortical surface with a micromanipulator, at least >0.5 mm away from the initial injection site. In the L5 injected group, some rats were imaged using a prism-like cannula inserted at a depth of 200–300 µm into the cortex (Fig. 1f). The craniotomy was then covered with a layer of silicon (Kwik-Cast™, World Precision Instruments, Inc.) and secured with dental cement. The animal was then placed in an arena of $40 \times 30 \times 20$ cm$^3$ (width, depth and height) with ad libitum food and water and connected to the setup via a flexible EEG/EMG recording cable (Pinnacle Technology, Inc.) and a fibre-optic patchcord (Doric Lenses) (Fig. 1a). Recording started typically after 1 to 2 h of recovery when the rats display normal waking EEG and behaviour (assessed by normal eating, drinking, grooming and alert exploration). A custom-build set-up was used for combined EEG/EMG and optical Ca$^{2+}$ recordings (Fig. 1a). Excitation light from a LED (450–490 nm, 50–70 µW) is relayed by a series of multimode fibre patch cords (Doric Lenses, diameters 400 µm (NA=0.37)) to the implanted cannula (diameters 400 µm). Emitted fluorescence is then relayed by the same series of fibres, deflected by a dichroic mirror (filter 500–700 nm) and the green light is detected by a photodiode (DET36A, Thorlabs, Dachau, Germany). Electrical signals (Ca$^{2+}$ and EEG/EMG) are then routed to an amplifier and collected by the commercially available sleep acquisition/analyses software VitalRecorder™ (Kissei Comtec, Irvine, CA, USA).

**EEG and fibre-optic Ca$^{2+}$ data processing and analysis.** EEG/EMG and Ca$^{2+}$ data were digitised at 200 Hz with a 0.5–100 Hz and a 0.1–30 Hz band-pass filters respectively. EMG was integrated using a 10–100 Hz band-pass filter. Offline, EEGs (FF and FP) and EMG signals were used to assign polygraphic data into 4 s epochs of AW, QW, IS, REM or NREM sleep (SleepSign for Animal; Kissei Comtec). Briefly, AW and QW were characterised by a high and variable EMG and desynchronised/low-amplitude EEG. AW was defined by the additional presence of high theta power (5–9 Hz) in the parietal EEG. NREM sleep was identified by low EMG, the presence of synchronised/high-amplitude EEGs and high sigma (9–16 Hz) activity. REM sleep displays the same EEG signature as AW but with no EMG activity typical of REM sleep muscle atonia. Finally, the IS was identified according to several criteria. A 4 s epoch was classified as IS if it presented a general increase in sigma activity and high theta power in the FP derivation as described in refs [23, 41, 42] (and see Supplementary Fig. 1). Because this EEG signature was quite common, we included additional criteria. An IS episode was defined as a sequence of at least 6 consecutive 4 s epochs and should follow a NREM sleep episode. Behavioural state scoring was done blind to the Ca$^{2+}$ signal. Percentage of total recording time and bout duration (>5 epochs[43]) for each vigilance state was calculated for the entire recording period. Fast Fourier transforms were performed on EEG and Ca$^{2+}$ signal for consecutive 4 s epochs. For each EEG, power was averaged within the slow oscillation (0.5–1.5 Hz), delta (1–4 Hz), theta (5–9 Hz), sigma (9–16 Hz), beta (16–30 Hz), slow gamma (Slow γ, 30–50 Hz) and fast gamma (Fast γ, 60–100 Hz) frequency bands. Ca$^{2+}$ activity changes were measured the same way using the average power in the 0.1–1 Hz frequency band (Fig. 2b and Supplementary Fig. 2a). To correct for interindividual differences and compare changes in EEG and Ca$^{2+}$ PD across behaviour states, all 4 s epoch PD values for a given frequency band (EEG and Ca$^{2+}$) were normalised to the mean of this particular frequency band across all behavioural states in each animal. The normalised changes in PD allowed comparing individuals while preserving the dynamic and magnitude of the changes observed. Those values were then expressed as trend by applying a moving average of a 24 s period every 4 s. Correlation analysis between Ca$^{2+}$ changes and EEG PD was done between 4 s epoch values.

**Episode third analysis.** Similar to previous published work[26], a behavioural episode was defined as a sequence of at least 13 epochs (≥52 s) of a given state, not interrupted by more than 30% of epochs of any other state. We identified 346 wake (AW+QW, mean duration: 271 ± 22.3 s), 548 SWS (mean duration: 235 ± 14.2 s) and 95 REM (mean duration: 114 ± 8.87 s) episodes. There was no difference in episode number (one-way ANOVA, Wake: $P = 0.724$; SWS: $P = 0.931$; REM: $P = 0.549$) and duration (one-way ANOVA, Wake: $P = 0.243$; SWS: $P = 0.419$; REM: $P = 0.278$) between groups for each state. Since episodes have different lengths, the analysis was performed as in ref. [26] by normalising the duration of each episode between 0 and 1 and subdividing this normalised duration into three "third" segments (1st, 2nd and 3rd). We then calculated the mean normalised PD (EEG and Ca$^{2+}$) within each third of individual behavioural state episodes. Those values were used to obtain the magnitude of PD changes within individual episodes, with ΔPD=PD in 3rd−PD in 1st (Fig. 4d).

**Transient detection and time–frequency analysis.** We developed a MATLAB-based software to perform additional analysis of EEG and Ca$^{2+}$ signals. After

extraction of the EEG and Ca$^{2+}$ signals, raw data (5 ms temporal resolution, sampling rate 200 Hz) were processed for Ca$^{2+}$ transient detection and time–frequency analysis.

The transient detection algorithm is a threshold algorithm using multiple pass. The signal was first normalised to obtain values between [0, 1] with the formula x_norm= $[x-\min(x)]/[\max(x)-\min(x)]$. The algorithm performed a series of passes (step of 0.1), searching first for transients with maximum amplitude (i.e., 1) down to the last pass that was defined by a minimal amplitude set by a threshold, here set at 0.2. The minimum and maximum transient durations were set at 0.5 and 6 s, respectively. Detections of transients <1 s were very rare and transients >6 s were represented by large signal fluctuations that included often more than one transient. In addition to Duration, the Amplitude of each transients was measured vertically from the lowest to the highest part of the transient.

The time–frequency analysis is based on the work of Lachaux et al.[28], adapted for continuous recordings and discrete frequency bands. For this analysis, we used the PD between 0.1 and 1 Hz for the Ca$^{2+}$ channel and all the frequency bands described above for the EEG channels. Briefly, the electrophysiological signals in all channels were processed with a moving search window that was set at 4 s. In the search window, the time–frequency transform (TF) of each channel is computed using a short-term Fourier transform. The TF is then sliced into overlapping (50%) subregions of interest (TFROI) of 500 ms. The mean "energy"[28] of each TFROI is then computed for each EEG channel except for the Ca$^{2+}$ channel where only the mean "energy" of TFROI at time 0 (T0) of the search window is computed. The search window is then time shifted by the TFROI time length (i.e., 500 ms) minus a 50% overlap (of the TFROI). This process is repeated until the end of the signal is reached. The result is, for each TFROI, a series of mean "energy" values. For every possible pair between TFROIs at T0 on the Ca$^{2+}$ channel and TFROIs on the EEG channels in the search window, Spearman's rank correlation coefficient is calculated using those series of mean energy values. We obtained a heatmap of correlation coefficient between each EEG channels and the Ca$^{2+}$ channel (Fig. 4c and Supplementary Fig. 5b). The x-axis of the heatmap is the time latency around T0 inside the search window on the EEG channels. The y-axis is the frequency bands chosen for the EEG channels.

**Pharmacology in anaesthetised animals.** To confirm that the optical signal recorded with the fibre-optic method reflected intracellular Ca$^{2+}$ changes, we used 8 additional rats in which we recorded dendritic activity under anaesthesia (surface cannula=4, prism cannula=4). After a 20 min EEG/Ca$^{2+}$ baseline recording, we applied 200 µl of Ni$^{2+}$ (2 µM, Sigma Aldrich)/Cd$^{2+}$ (1 µM, Santa Cruz Biotechnology) in rat ringer (135 mM NaCl, 5.4 mM KCl, 1.8 mM CaCl$_2$, 1 mM MgCl$_2$, 5 mM HEPES) on the surface of the cortex. The recording continued for additional 20 min (post-drug recording).

**Surgery for two-photon Ca$^{2+}$ imaging.** On the day of surgery, wild-type and Rbp4-cre mice (~P40) underwent surgery for EEG/EMG, virus injection, head-post and chronic window implantation (Supplementary Fig. 7a, b). The EEG/EMG surgery procedure was similar to the one for rats with slight adjustments. We used a custom-made EEG/EMG implant for FF and FP EEG recordings (Supplementary Fig. 7b). Stainless steel EEG and EMG wires were used. Electrodes were affixed to the skull with bone screws and dental acrylic. For chronic two-photon imaging, a 4 mm circular craniotomy was made on the left hemisphere above the barrel cortex (~1.5 mm posterior and 3.4 mm lateral of bregma). The dura was left intact. The injection procedure of GCaMP6s (AAV1.Syn.GCaMP6s.WPRE.SV40 in wild-type mice or AAV2/1-Syn-Flex-GCaMP6s-WPRE in the Rbp4-cre mouse, PENN Vector Core) into L5 (depth: 550 to 700 µm) was the same as in rats. After injection, the craniotomy was covered with a 4 mm glass coverslip (CS-4R, Warner Instruments, Hamden, CT, USA) and sealed with glue. A lightweight custom-made aluminium head-post was glued to the centre of the skull, between the EEG and window implant (Supplementary Fig. 7b). Finally, dental cement was used to cover the exposed skull and fixate the head-post and the EEG/EMG implant. At the end of the surgical procedure, buprenorphine was administered as a long-lasting analgesic (0.01 to 0.05 mg/kg IP) and the animals were allowed to recover for at least 3 days.

**Two-photon Ca$^{2+}$ imaging and data analysis.** Ca$^{2+}$ imaging sessions were performed between ZT0 and ZT12 (Supplementary Fig. 7a). Mice in the head-fixation stage were positioned underneath a resonant scanning two-photon microscope (B-Scope, Thorlabs, Newton, NJ, USA) equipped with GaAsP photomultiplier tubes (Hamamatsu, Tokyo, Japan). GCaMP6s was excited at 940 nm with a Ti:Sapphire laser (Mai-Tai DeepSee, Spectra-Physics, Santa Clara, CA, USA) and imaged through a 16×, 0.8 NA water immersion objective (Nikon, Tokyo, Japan). Full-frame images (512 × 512 pixels) were acquired capturing Ca$^{2+}$ activity. L5 cell bodies were imaged in three mice/depths between −450 and −600 µm (number of somata: 450 µm = 45; 500 µm = 22; 600 µm = 22) and apical shaft dendrites of L5 neurons were imaged in three mice/depths between −200 and −450 µm (number of dendrites: 200 µm = 47; 300 µm = 64; 450 µm = 24). For dendritic recordings, we followed dendrites down to L5 to control that they originate from the cell bodies in that layer. Single plane recordings of 4000 frames were continuously acquired over

1 to 2 h once the mouse started sleeping underneath the microscope. EEG signals were processed and analysed blindly to the $Ca^{2+}$ data in the same way as for rats.

Analysis of two-photon data was performed using ImageJ and a custom written software in MATLAB. ROIs were drawn by hand for each cell body and dendrite. For each ROI, pixel values inside the ROI were averaged to obtain the time series of $Ca^{2+}$ fluorescence. The raw fluorescence in each ROI was normalised using a 20 s sliding (5 ms) window on the continuous signal. Normalised fluorescence ($\Delta F/F_0$) was calculated as $(F-F_0)/F_0$, where $F_0$ is the mean lower third of the raw fluorescence values within the sliding window. $Ca^{2+}$ synchrony was calculated in 4 s epoch by calculating Pearson's correlation coefficient between all possible combinations of ROIs for a given field of view. An average of all Fisher transformed ("r-to-z") correlation coefficients was then made. The average underwent another Fisher transformation ("z-to-r") to obtain the synchrony level (normalised between 0 and 1).

**Juxtacellular recordings and data analysis**. Rats ($n = 2$, P37 on the day of surgery) were implanted under ketamine/xylazine anaesthesia (100 mg/kg, 5 mg/kg, IP) with a metal bolt for head fixation and a recording chamber (2 mm posterior and 2 mm lateral from bregma) for chronic access to hindlimb somatosensory cortex. Once the animal was habituated to sleep while head-fixed, daily sessions (over 3–4 days) of juxtacellular single-cell recordings of L5 neurons started. The recordings were performed at a mean depth reading of $1525 \pm 288$ μm ($n = 23$ cells). The glass pipette was filled with Ringer's solution containing NaCl 135, KCl 5.4, HEPES 5, $CaCl_2$ 1.8 and $MgCl_2$ 1 (pH 7.2). The juxtacellular signal was amplified and low-pass filtered at 3 kHz by a patch-clamp amplifier (Dagan, Minneapolis, MN, USA) and sampled at 25 kHz by a Power1401 data acquisition interface under the control of Spike2 software (CED, Cambridge, UK). State scoring was done blind to the firing pattern of cells using both LFP and EEGs signals. For cross-correlations, the EEG (FF and FP) and LFP signals was filtered for different frequency bands and cross-correlated with the instantaneous spike frequency of the recorded action potential train. To calculate the instantaneous spike frequency, the spike train was first converted into a modified sum of Dirac-delta functions, where the peak of each delta function was scaled to equal the acquisition frequency. This function was then convolved with a Gaussian function with s.d. of 20 ms (adapted from ref. [44]). For each frequency band, the peak (maximum within a 10 s window:±5 s) value for each cell was normalised to the s.d. We used visual detection of spindles which are easily identifiable in the LFP (unlike in the EEG). The onset and offset of a spindle was determined by the beginning and end of the train of spindle oscillations which was often delimited by distinct UP states. A total of 476 spindles were detected across the 23 recordings. To measure the correlation between EEG/LFP and spindle density we used recordings of 17 out of the 23 cells as some recordings showed a drift in the LFP signal that could have biased the results. Recordings were broken up into 10 s consecutive windows with an overlap of 0.25 s between successive windows. Within each window, we calculated the spindle density and the EEG and LFP power for individual frequency bands (see above). Spindle density was measured as the cumulative duration of detected spindles during the 10 s window. For each recording and each frequency band, a scatter plot was generated plotting the total power of the frequency band against the spindle density of each 10 s window. A linear regression was performed on the scatter plot and the correlation was measures as the size of the slope factor.

**Habituation to head fixation for rats and mice**. Following surgery, mice and rats were trained to naturally sleep while being head-fixed. On the first day, animals were allowed to freely explore the head-fixation stage. Over the next days, the duration of head fixation was increased daily by 5, 15, 30 and 60 min to minimise stress. At the beginning and end of each session animals received condensed milk as reward. During head fixation, EEG/EMG was recorded to reveal naturally occurring periods of sleep. Mice usually started sleeping occasionally after 7 days of training and displayed consolidated sleep episodes after 2 to 3 weeks of surgery, when expression levels of GCaMP6s were also sufficient. Rats express consolidated sleep after only 1 week of training.

**Brain slicing and imaging**. Images from the injection sites were obtained from brain slices as previously described[16]. Briefly, after killing, the brain was rapidly removed into ice-cold, oxygenated artificial cerebrospinal fluid containing (in mM): 125 NaCl, 25 $NaHCO_3$, 2.5 KCl, 1.25 $NaH_2PO_4$, 1 $MgCl_2$, 25 glucose and 2 $CaCl_2$ (pH 7.4). Slices (300 μm) were cut with a vibrating microslicer on a block angled at 15° to horizontal and maintained at 37 °C in the preceding solution for 30 min before use. The fluorescence signal was obtained using an LED light source (CoolLED, 480 nm), standard epifluorescence filter sets for FITC used for OGB-1 AM and GCaMP6s and a CoolSNAP EZ CCD camera (Photometrics).

**Statistics**. All statistics were calculated using a commercial software (SigmaStat, Systat Software Inc., San Jose, CA, USA). All data were tested for normality and equal variance. Parametric data were assessed using Student's t-tests for planned, single comparisons or one-, two- or three-way ANOVA and Holm–Sidak test for multiple post hoc comparisons. In cases where nonparametric statistics were required, Mann–Whitney rank sum tests were used for planned, single

comparisons and Kruskal–Wallis one- or two-way ANOVA and Dunn's tests for multiple post hoc comparisons. Correlations were calculated using Pearson's correlation coefficient.

**Code availability**. All data codes are available from the corresponding authors on request.

**Data availability**. All data are available from the corresponding authors on request.

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

## Acknowledgements

We thank Uwe Heinemann, Igor Timofeev and Raphaelle Winsky-Sommerer for their helpful comments on the manuscript. and Derk-Jan Dijk and Sandro Lecci for helpful discussions. Fabian Bentley and Jessica Unger for technical assistance and Benjamin Judkewitz for help with optimisation of our one-photon detection system. This research was supported by the Swiss National Science Foundation (31003A_130694) grant, DFG (EXC 257 NeuroCure) to M.E.L. and FP7 Marie-Curie Actions IRG grant (268273) to J.S.

## Author contributions

J.S. designed the experiments with input from all authors on this paper. J.S., D.d.L. and C.J.R. built the one-photon recording set-up. Freely behaving experiments were carried out and analysed by J.S. Two-photon experiments and analysis were carried out by J.S.-G. and J.S. with help from N.T. Analysis software for fibre-optic and two-photon data were developed by C.J.R. with input from N.T. for two-photon data. Juxtacellular recordings were carried out by G.D. and J.S. and analysed by D.I.K. and J.S. with input from G.D. C.B. performed brain slicing and imaging. J.S. and M.E.L. wrote the manuscript with input from all authors on this paper.

## Additional information

**Competing interests:** The authors declare no competing financial interests.

**Change history:** A correction to this article has been published and is linked from the HTML version of this paper.

