## [Peer Review File · Nature Communications]

Reviewers' comments:

Reviewer #1 (Remarks to the Author):

In this paper by Seibt et al., the authors performed simultaneous EEG and calcium (Ca²⁺) recordings from the dendrites of layer 5 (L5) cortical pyramidal neurons and report that spindle-like oscillations are important in promoting synchronization of Ca²⁺ activity in L5 dendrites. The authors' speculate that this, in turn, may "prime structural plasticity consolidation occurring during REM sleep". This is a technically ambitious paper that addresses an important knowledge gap in the experimental neurosciences. I did have two concerns, one of which was technically, the other of which was conceptual.

My major technical concern is as follows:

The authors used two methods for loading the Ca²⁺ indicator: in one case they soaked the cortical region and the Ca²⁺ indicator was picked up by neurons: this is a low yield and non-selective method. In the other case they used a viral approach, which is higher yield, but also non-selective. Using these approaches, were the authors able to somehow follow the specific dendrites that originate from the cell bodies from the pyramidal neurons in L5? This was not clear to me, but being able to do so is central to the very specific claims that the authors are making.

My major conceptual concerns is as follows:

Although the authors provide compelling evidence that spindle-like oscillations are important in promoting synchronization of Ca²⁺ activity in dendrites of L5 neurons, they do not provide demonstrate physiologic/neurobiologic relevance. In other words, is Ca²⁺ activity in dendrites of L5 neurons during natural sleep necessary for the cortical plasticity underlying various cognitive functions? I feel that this is an important missing element of an otherwise nicely executed study, as it provides an element of causality that is currently lacking.

Reviewer #2 (Remarks to the Author):

Using simultaneous calcium activity and EEG measurement, Seibt et al. identify layer and subcellular compartment specific relationships in calcium responses with ongoing oscillations. They observed that L5 dendrites show specific increased responses and synchrony during spindle oscillations during sleep. The authors suggest that this synchrony reflect or enable local dendritic plasticity during sleep periods. These are a novel and interesting set of findings that represent a potential link between brain states and mechanisms of synaptic plasticity.

I have a few comments and points of clarification.

1) For fiber-optic and 2-photon Ca²⁺ measures 'activity' is represented by two different analysis methods (0.1-1Hz spectral power vs $\Delta F/F_0$). This presents some points of confusion in understanding the findings from these different modes of measurement. There are some inconsistencies in the results and their interpretation which might stem from if the interpretation was based off of fiber optic or 2P measurements:

a) For example:, in the result section is stated that dendritic Ca²⁺ does not increase with REM sleep. This is true for population fiber-optic measures. However, not for 2-photon measures of single dendrites.

b) In the introduction it is stated that L5 soma do not show the same findings than L5 dendrites. However, in the 2P data, L5 soma show similar trends to dendrites (see Fig.5)

c) Result section title about Figure 6 states that 'Ca²⁺ activity is synchronized in population of dendrites, but not cell bodies, during SWS.' Yet, as shown in Fig.6, L5 soma showed overall stronger synchronization than dendrites, also in SWS. It is more that changes in synchronization level is particularly correlated with sigma-beta power during SWS in dendrites. L5 soma is also correlated with sigma-beta power, however also similarly with other frequency ranges (less specific).

In general, it might be helpful to understand the exact relationship between the PD measurements from the fiber optic recordings and activity across multiple individual dendrites from 2P imaging. The authors should be able to take the 2P imaging and extract a PD measurement by combining all the individual dendrites in their FOV.

2) Where is the power spectrum of EEG signals shown? Is there a spectral power peak in the EEG signal in the sigma range? Authors should explicitly define their rationale of frequency-selection and filtering the data in certain ranges.

3) In the introduction it is stated that L5- pyramidal neurons contribute dominantly to the EEG signal. Even though this is a plausible assumption, a recent study (Musall et al. (2014)) has shown that spatial neuronal correlations also shape critically EEG signals. This should be mentioned when discussing the relationship between dendritic/soma spatial synchrony and EEG components.

4) Fig.3B seems to indicate that there is correlation between L2/3 soma and sigma-band, even though it is much weaker than L5 dendrites. Is this significant?

5) The finding that L5 soma spiking activity is decoupled from EEG fluctuations during SWS is a really surprising finding. In Figure title is states that 'Spiking activity of L5 cell bodies is not influenced by spindles'. In fact, shown in Fig7E, L5 spiking is not influenced by any band of the EEG (hence more general), which is surprising and interesting. (A) It would be nice if the authors can elaborate on this more, including methodological considerations. (B) Is there a decoupling also with the LFP from juxtacellular recordings? How is the relation between LFP and EEG? (C) Is the decoupling specific to SWS or general?

6) Typo: Figure 4. Local changes IN sigma-beta power correlates with changes in dendritic activity during SWS.

7) Typo: Figure 5 Spindle-beta oscillations reflect increased and DECREASED Ca²⁺ activity in single dendrites

Reviewer #3 (Remarks to the Author):

This is a very interesting paper with truly novel and very exciting data on calcium signaling in cortical dendrites during sleep. This paper is the first to recognize that sleep EEG patterns are accompanied by distinct oscillatory calcium signaling in apical dendritic trees. The calcium oscillations are observed in layer 5 dendrites but not in layer 2/3 or layer 5 cell somata. Even more intriguingly, the strength of this calcium oscillation is temporally best correlated with the power fluctuations in the sigma (9-16 Hz) frequency band. This correlation extends to the single dendrite level yet is not reflected in the action potential discharge pattern of layer V neurons. This suggests that, through some yet undefined mechanisms, sigma power activity produces a slowly fluctuating cortical state in which layer V dendrites are periodically decoupled from the soma.

This paper will be of tremendous importance to understand brain activity during sleep. My impression is that it will completely re-shape the view of what sleep rhythms do to the cortex. In the case of spindles, instead of simply "relaying" the 10-15 Hz frequencies generated in thalamus, activity is somehow integrated to generate slow fluctuations of sigma power and associated calcium oscillations. This study will also be key to further understand how sleep promotes memory, and why sleep leads to disconnection from the environment.

In addition to its scientific relevance, the study excels by a combination of techniques that resolve cellular and population dendritic calcium signals, single unit activity and EEG. Data presentation and statistics appear sound.

Here are some suggestions for further improvement:

- 1) It is not clear when sleep recordings were actually done during the day and what the time period is over which the data in 1e were actually calculated? Why were "only" 2.5 hrs considered in these freely moving animals that can sleep whenever they want? Were the time-of-day values the same for all animals?
- 2) How the state of IS is determined is not very clear. The duration value provided is given with a precision that seems exaggerated compared to the indications in the raw data. The onset of IS sometimes coincides with a period with still little calcium activity, sometimes in the middle of a calcium peak (Fig 2d). According to Gottesmann, theta bouts start appearing during IS. Perhaps this increase in theta power could be seen in the parietal EEG? If this is not possible, strict duration indications should be replaced by approximate ranges.
- 3) The behavioral state classifications are quite arbitrary and do not follow standard rules. Microarousals can be shorter than 5 epochs and they will invariably disturb a consolidated bout. Moreover, they are frequent, in particular during head-restrained conditions. Therefore, classifying fragmented sleep into consolidated sleep bears the risk of inducing power fluctuations that can then affect the apparent time course of power values across

apparently consolidated NREM sleep bouts.

4) It is not clear how NREM sigma power relates to the calcium PD during REM sleep in 3c. At what moment during NREM sleep was sigma power calculated? Talking about sigma power during REM sleep is likely going to be very confusing and should be put into the supplemental materials.

5) It could be assessed whether or not exits to different vigilance states from IS were in any way different in terms of IS parameters.

6) Ca signals were analyzed with respect to their frequency compositions using FFT. It is not clear how the data presented in 2b were obtained. First, the data suggest that the 0.1-1Hz component in the calcium signals in the dendrites is present in all vigilance states. However AW, QW, and also REM signals seem not to contain much of this (see 2a and 2d). Could the peak obtained be an artefact of doing the FFT over too short stretches of the behavioral states? Please provide the distribution of vigilance state bout durations and show that the FFT amplitude does not depend on the bout lengths included. Second, how was the FFT constructed given that its resolution will vary depending on bout length? Third, what do the individual data points represent in 2b and how was this binning done? Fourth, how are statistics done over a number of datapoints in the FFT that exceeds the number of animals used?

7) The parallel increase of delta and sigma power during a NREM sleep bout is very surprising. It contradicts the widely documented notion that delta and sigma are anticorrelated during sleep. One reason could be that sleep was not measured at the same times of day, which could result in widely different delta power values. As also sleep bouts shorten with time of days, this could additionally lead to apparent increases. One idea would be to restrict this analysis to only certain periods of the day and to normalize the data only for each vigilance state.

8) On top of the 0.1-1Hz fluctuations, the data in Fig3 suggest that there is an additional infra-slow fluctuation over which both Ca and sigma PD vary. This is strikingly reminiscent of a report recently published (Lecci et al., Science Advances, February 2017) that reports on a 0.02-Hz oscillation in sigma power that is anticorrelated with heart rate and that determines behavioral arousability. It would be interesting to determine whether the 0.02 Hz-frequency component is also present in this study.

9) The authors should apply some caution in equating sigma power with spindles. It would need to be shown directly that increased sigma power goes along with elevated spindle density, and spindles would need to be counted.

10) Any idea of how the 9-16 Hz rhythmic inputs from thalamus are transformed into 0.1-1 Hz calcium oscillations? In the non-filtered calcium signal, is there any evidence for faster phasic events that could reflect individual spindles? Alternatively, should we start thinking that spindle-related thalamocortical EPSPs are temporally integrated such that the original spindle frequency disappears and that dendrites become disinhibited over a time scale 1-2 times slower than the spindle events? Could this be a reason that there is still much controversy going on about how to define cortical spindles?

11) Why was the 2P imaging not done over hindlimb area in mice?

Reviewers' comments:

We would like to thank all the reviewers for their helpful comments and suggestions that have, we think, significantly improve the manuscript.

Reviewer #1 (Remarks to the Author):

In this paper by Seibt et al., the authors performed simultaneous EEG and calcium (Ca²⁺) recordings from the dendrites of layer 5 (L5) cortical pyramidal neurons and report that spindle-like oscillations are important in promoting synchronization of Ca²⁺ activity in L5 dendrites. The authors' speculate that this, in turn, may "prime structural plasticity consolidation occurring during REM sleep". This is a technically ambitious paper that addresses an important knowledge gap in the experimental neurosciences. I did have two concerns, one of which was technically, the other of which was conceptual.

My major technical concern is as follows:

The authors used two methods for loading the Ca²⁺ indicator: in one case they soaked the cortical region and the Ca²⁺ indicator was picked up by neurons: this is a low yield and non-selective method. In the other case they used a viral approach, which is higher yield, but also non-selective. Using these approaches, were the authors able to somehow follow the specific dendrites that originate from the cell bodies from the pyramidal neurons in L5? This was not clear to me, but being able to do so is central to the very specific claims that the authors are making.

Thank you for pointing out this issue. The question of specificity to dendrites of L5 pyramidal neurons was addressed most thoroughly in our original paper using this method (Murayama et al., 2007). Since that publication, we have used the method many times and have controlled for this question in each study (Murayama and Larkum, 2009a, 2009b; Murphy et al., 2016). In this study, we were also able to control for this question. Firstly, we performed a subset of dendritic fibre-optic recordings using a prism to shine light at 90° in L2/3, ensuring that the illumination light was cast only on the dendrites of L5 neurons (Fig. 1f and Supplementary Fig.3). Secondly, using the two-photon approach we could directly control which elements in L2/3 were fluorescing after injection of the virus to L5. Here, we also followed the dendrites from the upper layers down to their cell bodies which confirmed (once again) the specificity and efficacy of our labelling method. Finally, we found equivalent results using a synthetic and viral expressed Ca²⁺ indicators (cf. Fig. 2 in (Murayama et al., 2007) with our new Figure 1b, previously Supplementary Fig.1)

Although the authors provide compelling evidence that spindle-like oscillations are important in promoting synchronization of Ca²⁺ activity in dendrites of L5 neurons, they do not provide demonstrate physiologic/neurobiologic relevance. In other words, is Ca²⁺ activity in dendrites of L5 neurons during natural sleep necessary for the cortical plasticity underlying

various cognitive functions? I feel that this is an important missing element of an otherwise nicely executed study, as it provides an element of causality that is currently lacking.

We agree with the reviewer that investigating the causal relationship between dendritic Ca^{2+} activity, spindles and cortical plasticity is an important next step. We also feel it is out of the scope of this study for the following reasons: it will undoubtedly require a complete set of new experiments that will be technically challenging, will take several years and probably need as many figures as we already have in the current paper. To carry out this study properly we would first need to establish a technique for suppressing dendritic activity without side effects. We would need to devise a control-loop system to invoke this suppression during spindle activity and finally we would need to carry out this suppression in a suitable behavioural paradigm involving cortical plasticity with a quantifiable outcome. We are nevertheless very keen to get on with the study the reviewer suggests.

References:

Murayama, M., and Larkum, M.E. (2009a). In vivo dendritic calcium imaging with a fiberoptic periscope system. *Nat. Protoc.* 4, 1551–1559.

Murayama, M., and Larkum, M.E. (2009b). Enhanced dendritic activity in awake rats. *Proc. Natl. Acad. Sci. U. S. A.* 106, 20482–20486.

Murayama, M., Pérez-Garci, E., Lüscher, H.-R., and Larkum, M.E. (2007). Fiberoptic system for recording dendritic calcium signals in layer 5 neocortical pyramidal cells in freely moving rats. *J. Neurophysiol.* 98, 1791–1805.

Murphy, S.C., Palmer, L.M., Nyffeler, T., Müri, R.M., and Larkum, M.E. (2016). Transcranial magnetic stimulation (TMS) inhibits cortical dendrites. *eLife* 5.

Reviewer #2 (Remarks to the Author):

Using simultaneous calcium activity and EEG measurement, Seibt et al. identify layer and subcellular compartment specific relationships in calcium responses with ongoing oscillations. They observed that L5 dendrites show specific increased responses and synchrony during spindle oscillations during sleep. The authors suggest that this synchrony reflect or enable local dendritic plasticity during sleep periods. These are a novel and interesting set of findings that represent a potential link between brain states and mechanisms of synaptic plasticity.

I have a few comments and points of clarification.

1) For fibre-optic and two-photon Ca²⁺ measures ‘activity’ is represented by two different analysis methods (0.1-1Hz spectral power vs $\Delta F/F_0$). This presents some points of confusion in understanding the findings from these different modes of measurement. There are some inconsistencies in the results and their interpretation which might stem from if the interpretation was based off of fibre optic or 2P measurements:

- a) For example:, in the result section is stated that dendritic Ca²⁺ does not increase with REM sleep. This is true for population fibre-optic measures. However, not for two-photon measures of single dendrites.

We do not think that our fibre-optic and two-photon data are inconsistent. The apparent discrepancy comes from the fact that the fibre-optic method is more sensitive to changes at the population level, i.e. synchronized responses, whereas single-cell data is insensitive to this aspect. An analogy to the REM sleep situation is the “party effect” where the ambient noise in the room (i.e. population recordings) appears not to fluctuate whereas the noise level from any individual can be fluctuating substantially. On the other hand, if the fluctuations are synchronous (e.g. SWS), this is clearly detectable at the population level. In fact, the two-photon method can be applied to single cells and to a population (as the reviewer points out below in question c) and so we could use the same two-photon data to demonstrate this point: single dendrite two-photon recordings show fluctuations during REM sleep that do not manifest in two-photon recordings at the population level (see Fig. II below). Furthermore, our data during SWS show large responses with all approaches consistent with this explanation. We now discuss this apparent discrepancy for the REM sleep results in our revised manuscript as we think that it is of general interest when interpreting and comparing results from bulk/population and two-photon/single cell imaging, particularly in relation to brain states when network synchronisation prevails.

- b) In the introduction it is stated that L5 soma do not show the same findings than L5 dendrites. However, in the 2P data, L5 soma show similar trends to dendrites (see Fig.5)

We appreciate the reviewer pointing this out. We hypothesized that the L5 soma data is contaminated with responses from both L5 pyramidal and nearby inhibitory neurons. This is plausible since it has been shown that inhibitory neurons in the cortex increase their activity during spindles in rodents (Peyrache et al., 2011). We investigated this question with a control experiment by using a transgenic mouse line expressing GCaMP6s specifically in L5

pyramidal neurons (i.e. with no contamination from inhibitory neurons). Here, we found that correlation for individual soma were generally much weaker ($r < \pm 0.1$, compare to Fig. 5d) and did not show a similar trend compared to dendrites (cf. Fig. 5f and Fig. I below). We now included this in the discussion of the paper.

Figure I. Correlation between Ca^{2+} activity in individual soma of L5 pyramidal cell and EEG. For this experiment, the Rbp4-cre mouse line was used to express GCaMP6s (floxed-AAV) specifically in L5 pyramidal neurons. *Left panel.* Correlation between $\Delta F/F_0$ for all single soma ($n = 41$) and EEG-FP PDs for different frequency bands during SWS (slow oscillations and delta bands were combined into SWA due to similar trends). Values from individual 4-s SWS epochs were used for correlation. *Right panel.* Mean correlation across somata selected to have a positive ($r > 0$) or negative ($r < 0$) correlation with sigma PD (compare results to Fig.5f).

c) Result section title about Figure 6 states that ‘ Ca^{2+} activity is synchronized in population of dendrites, but not cell bodies, during SWS.’ Yet, as shown in Fig.6, L5 soma showed overall stronger synchronization than dendrites, also in SWS. It is more that changes in synchronization level is particularly correlated with sigma-beta power during SWS in dendrites. L5 soma is also correlated with sigma-beta power, however also similarly with other frequency ranges (less specific).

We agree that the section title does not reflect appropriately the results shown in Figure 6 as L5 somata synchronisation is also elevated during SWS. We have changed the title to “*Dendritic activity synchronisation during sigma-beta oscillations*” and the wording within the text to highlight the idea of “specificity” of correlation of dendritic compared to somatic synchronisation with sigma-beta oscillations.

In general, it might be helpful to understand the exact relationship between the PD measurements from the fibre optic recordings and activity across multiple individual dendrites from 2P imaging. The authors should be able to take the 2P imaging and extract a PD measurement by combining all the individual dendrites in their FOV.

We agree with the reviewer that a similar PD analysis of population activity extracted from our two-photon data is important to compare results with the population activity measured with the fibre-optic approach. To replicate the exact same analysis as for our fibre-optic recordings, we have calculated the average $\Delta F/F_0$ for each of the FOVs ($n = 3$ mice) used for two-photon dendritic Ca^{2+} imaging. The average FFT power spectra and PD (0.1-1Hz) of the

Ca²⁺ signal was then calculated for each behavioural state (see Methods and Fig. II below). The reviewer can see that the power spectra (peak at frequencies < 1Hz) and distribution of PD across states show a similar trend as for our fibre-optic recordings of dendritic Ca²⁺ activity. Notably, we confirm a lower level of dendritic activity during REM sleep at the population compared to single dendrite level (cf Fig. 2c), as pointed out by the reviewer in question a). The changes in Ca²⁺ PD and Ca²⁺ activity synchronisation derived from our two-photon measurements are actually quite similar (compare Ca²⁺ PD changes in Figure I with Fig. 6b). This further supports our interpretation that activity synchronisation might be an important component that reflects population activity measures. We decided not to include these data in the revised paper as we feel that more animals are needed to meet the standard for publication. But we are happy to include the figure as Supplementary material as per

Figure II. Two-photon PD analysis. *Left panel.* Average FFT power spectra of Ca²⁺ activity in population of dendrites (mean $\Delta F/F_0$ within FOV) across animals and behavioural states. *Right panel.* Average (grey bars) and single (black circles) PD (0.1-1Hz) values across behavioural states. N = 3 mice. Values for individual mice are not represented on the power spectra for clarity but reported for the Ca²⁺ PD results.

request.

2) Where is the power spectrum of EEG signals shown? Is there a spectral power peak in the EEG signal in the sigma range? Authors should explicitly define their rationale of frequency-selection and filtering the data in certain ranges.

We thank the reviewer for giving us the opportunity to clarify this question. While surface EEG recordings in humans can reveal a specific peak around sigma power (more evident in frontal regions), this peak is not found in rodent EEG (see figure 2 in (Mölle et al., 2009) for a comparison between human and rat NREM power spectra). The EEG power spectra are now presented as new Supplementary material (Supplementary Fig. 1a) and show standard profiles across behavioural states in rodents (e.g. peak in the theta band during active wake, IS and REM sleep in the parietal EEG and high delta activity during NREM sleep in both derivations).

More generally, our rationale for frequency range selection, in particular sigma (9-16Hz), is based on an extensive review of the literature to ensure that our frequency bands meet the standards in rodent, especially rat, sleep studies (selection of reference papers: (Astori et al., 2013; Benington et al., 1994; Bjorvatn et al., 1998; Buzsáki et al., 2013; Carr et al., 2012;

Colgin, 2013; Crunelli and Hughes, 2010; Düzel et al., 2010; Franken et al., 1991, 1998, Gottesmann, 1992, 1996; Grosmark et al., 2012; Lecci et al., 2017; Neckelmann et al., 1994; Robert et al., 1999; Schwierin et al., 1999; Terrier and Gottesmann, 1978; Trachsel et al., 1988; Vyazovskiy et al., 2009; Watson and Buzsáki, 2015; Watson et al., 2016)). We concede that the definition of frequency bands can slightly vary between rodent studies but this is an intrinsic problem in the field of sleep, and more generally neural oscillations, that is difficult to avoid.

3) In the introduction it is stated that L5- pyramidal neurons contribute dominantly to the EEG signal. Even though this is a plausible assumption, a recent study (Musall et al. (2014)) has shown that spatial neuronal correlations also shape critically EEG signals. This should be mentioned when discussing the relationship between dendritic/soma spatial synchrony and EEG components.

We thank the reviewer for bringing this paper to our attention. This paper is particularly relevant in the context of our study as our main results suggest that dendritic activity synchrony reflects cortical spindle EEG. We have included this study in our discussion.

4) Fig.3B seems to indicate that there is correlation between L2/3 soma and sigma-band, even though it is much weaker than L5 dendrites. Is this significant?

If with significant, the reviewer refers to statistical significance of the correlations, then yes (note: individual L2/3 recordings displayed both negative and positive correlations while L5 revealed only highly significant positive correlations). However, the weakness of L2/3 somata correlations compared to dendritic correlations leads us to think that it is not very important and also because this correlation was not specific to the sigma band when compared to other frequency bands (see Supplementary Fig. 5a).

5) The finding that L5 soma spiking activity is decoupled from EEG fluctuations during SWS is a really surprising finding. In Figure title it states that 'Spiking activity of L5 cell bodies is not influenced by spindles'. In fact, shown in Fig7E, L5 spiking is not influenced by any band of the EEG (hence more general), which is surprising and interesting.

We agree with the reviewer that our result showing a decoupling of L5 somatic spiking with EEG fluctuations was surprising. We have now performed a more controlled analysis, including both EEGs (frontal and parietal) and LFP (as requested by the reviewer). Our new results are presented in panel e) in Figure 7.

(A) It would be nice if the authors can elaborate on this more, including methodological considerations.

The methodology for the cross-correlation analysis is described in detail in the Methods section. For each cell, we included a control cross-correlation validation step (i.e. between the delta band and the raw EEGs and LFP signals) to ensure that our results were not affected by the quality of the signal.

(B) Is there a decoupling also with the LFP from juxtacellular recordings? How is the relation between LFP and EEG?

The firing of L5 cells measured with juxtacellular recordings was decoupled with the LFP except in the delta frequency band. We did not investigate in detail the relation between EEG and LFP as we did not know what specific aspect the reviewer was curious about. However, EEG and LFP reflect neuronal network activity changes at different scales (global vs. local, respectively) and oscillatory fluctuations extracted from both measures should, in theory, provide similar information. Our new results demonstrate that in relation to L5 neuronal spiking and spindle density, correlation analysis shows similar trends with both LFP and EEG (Fig. 7e,f).

(C) Is the decoupling specific to SWS or general?

Since our Ca^{2+} imaging results show a specific relation between dendritic activity of L5 neurons and sigma-beta EEG fluctuations, our analysis for this paper focused mainly on SWS. Ongoing experiments in the lab are investigating the relation between cortical cell spiking and EEG/LFP fluctuations across different brain states and cell types (i.e. excitatory vs. inhibitory) in a more detailed manner. Results from these experiments will be included as part of a separate study.

6) Typo: Figure 4. Local changes IN sigma-beta power correlates with changes in dendritic activity during SWS.

Corrected.

7) Typo: Figure 5 Spindle-beta oscillations reflect increased and DECREASED Ca^{2+} activity in single dendrites

Corrected.

References:

- Astori, S., Wimmer, R.D., and Lüthi, A. (2013). Manipulating sleep spindles--expanding views on sleep, memory, and disease. *Trends Neurosci.* 36, 738–748.
- Benington, J.H., Kodali, S.K., and Heller, H.C. (1994). Scoring transitions to REM sleep in rats based on the EEG phenomena of pre-REM sleep: an improved analysis of sleep structure. *Sleep* 17, 28–36.
- Bjorvatn, B., Fagerland, S., and Ursin, R. (1998). EEG power densities (0.5-20 Hz) in different sleep-wake stages in rats. *Physiol. Behav.* 63, 413–417.
- Buzsáki, G., Logothetis, N., and Singer, W. (2013). Scaling brain size, keeping timing: evolutionary preservation of brain rhythms. *Neuron* 80, 751–764.
- Carr, M.F., Karlsson, M.P., and Frank, L.M. (2012). Transient slow gamma synchrony underlies hippocampal memory replay. *Neuron* 75, 700–713.
- Colgin, L.L. (2013). Mechanisms and functions of theta rhythms. *Annu. Rev. Neurosci.* 36, 295–312.
- Crunelli, V., and Hughes, S.W. (2010). The slow (<1 Hz) rhythm of non-REM sleep: a dialogue between three cardinal oscillators. *Nat. Neurosci.* 13, 9–17.
- Düzel, E., Penny, W.D., and Burgess, N. (2010). Brain oscillations and memory. *Curr. Opin. Neurobiol.* 20, 143–149.

- Franken, P., Dijk, D.J., Tobler, I., and Borbély, A.A. (1991). Sleep deprivation in rats: effects on EEG power spectra, vigilance states, and cortical temperature. *Am. J. Physiol.* 261, R198-208.
- Franken, P., Malafosse, A., and Tafti, M. (1998). Genetic variation in EEG activity during sleep in inbred mice. *Am. J. Physiol.* 275, R1127-1137.
- Gottesmann, C. (1992). Detection of seven sleep-waking stages in the rat. *Neurosci. Biobehav. Rev.* 16, 31–38.
- Gottesmann, C. (1996). The transition from slow-wave sleep to paradoxical sleep: evolving facts and concepts of the neurophysiological processes underlying the intermediate stage of sleep. *Neurosci. Biobehav. Rev.* 20, 367–387.
- Grosmark, A.D., Mizuseki, K., Pastalkova, E., Diba, K., and Buzsáki, G. (2012). REM sleep reorganizes hippocampal excitability. *Neuron* 75, 1001–1007.
- Lecci, S., Fernandez, L.M.J., Weber, F.D., Cardis, R., Chatton, J.-Y., Born, J., and Lüthi, A. (2017). Coordinated infraslow neural and cardiac oscillations mark fragility and offline periods in mammalian sleep. *Sci. Adv.* 3, e1602026.
- Mölle, M., Eschenko, O., Gais, S., Sara, S.J., and Born, J. (2009). The influence of learning on sleep slow oscillations and associated spindles and ripples in humans and rats. *Eur. J. Neurosci.* 29, 1071–1081.
- Neckelmann, D., Olsen, O.E., Fagerland, S., and Ursin, R. (1994). The reliability and functional validity of visual and semiautomatic sleep/wake scoring in the Møll-Wistar rat. *Sleep* 17, 120–131.
- Peyrache, A., Battaglia, F.P., and Destexhe, A. (2011). Inhibition recruitment in prefrontal cortex during sleep spindles and gating of hippocampal inputs. *Proc. Natl. Acad. Sci. U. S. A.* 108, 17207–17212.
- Robert, C., Guilpin, C., and Limoge, A. (1999). Automated sleep staging systems in rats. *J. Neurosci. Methods* 88, 111–122.
- Schwierin, B., Achermann, P., Deboer, T., Oleksenko, A., Borbély, A.A., and Tobler, I. (1999). Regional differences in the dynamics of the cortical EEG in the rat after sleep deprivation. *Clin. Neurophysiol. Off. J. Int. Fed. Clin. Neurophysiol.* 110, 869–875.
- Terrier, G., and Gottesmann, C.L. (1978). Study of cortical spindles during sleep in the rat. *Brain Res. Bull.* 3, 701–706.
- Trachsel, L., Tobler, I., and Borbély, A.A. (1988). Electroencephalogram analysis of non-rapid eye movement sleep in rats. *Am. J. Physiol.* 255, R27-37.
- Vyazovskiy, V.V., Olcese, U., Lazimy, Y.M., Faraguna, U., Esser, S.K., Williams, J.C., Cirelli, C., and Tononi, G. (2009). Cortical firing and sleep homeostasis. *Neuron* 63, 865–878.
- Watson, B.O., and Buzsáki, G. (2015). Sleep, Memory & Brain Rhythms. *Daedalus* 144, 67–82.
- Watson, B.O., Levenstein, D., Greene, J.P., Gelinas, J.N., and Buzsáki, G. (2016). Network Homeostasis and State Dynamics of Neocortical Sleep. *Neuron*.

Reviewer #3 (Remarks to the Author):

This is a very interesting paper with truly novel and very exciting data on calcium signaling in cortical dendrites during sleep. This paper is the first to recognize that sleep EEG patterns are accompanied by distinct oscillatory calcium signaling in apical dendritic trees. The calcium oscillations are observed in layer 5 dendrites but not in layer 2/3 or layer 5 cell somata. Even more intriguingly, the strength of this calcium oscillation is temporally best correlated with the power fluctuations in the sigma (9-16 Hz) frequency band. This correlation extends to the single dendrite level yet is not reflected in the action potential discharge pattern of layer V neurons. This suggests that, through some yet undefined mechanisms, sigma power activity produces a slowly fluctuating cortical state in which layer V dendrites are periodically decoupled from the soma.

This paper will be of tremendous importance to understand brain activity during sleep. My impression is that it will completely re-shape the view of what sleep rhythms do to the cortex. In the case of spindles, instead of simply “relaying” the 10-15 Hz frequencies generated in thalamus, activity is somehow integrated to generate slow fluctuations of sigma power and associated calcium oscillations. This study will also be key to further understand how sleep promotes memory, and why sleep leads to disconnection from the environment.

In addition to its scientific relevance, the study excels by a combination of techniques that resolve cellular and population dendritic calcium signals, single unit activity and EEG. Data presentation and statistics appear sound.

We thank the reviewer for his/her appreciation of the significance of our work.

Here are some suggestions for further improvement:

1) It is not clear when sleep recordings were actually done during the day and what the time period is over which the data in 1e were actually calculated? Why were “only” 2.5 hrs considered in these freely moving animals that can sleep whenever they want? Were the time-of-day values the same for all animals?

As shown in our experimental design (now Fig. 1c, previously Supplementary Fig. 1a), all our freely behaving recordings were performed at the same period during the light phase (between ZT6-ZT12). The duration of recordings used for further analysis were restricted to ~2-hrs for several methodological reasons. Firstly, recordings using OGB1-AM as Ca^{2+} indicator is intrinsically limited due to the time-course of bleaching of this dye (after ~5-hrs). Secondly, while recordings using genetically encoded virus (i.e. GCaMP6s) could be extended, for comparison purposes we performed the recordings under similar conditions as for OGB1-AM. Finally, to ascertain that the data used for analysis were of the best quality possible, we restricted our analyses to the early periods of Ca^{2+} signal to avoid contamination of signal drifts that often occur after several hours of continuous recordings in freely behaving animals. The reasons for this drift is not clear but could be due to a combination of several factors (e.g. dye bleaching, cannula/brain movements, and/or brain tissue scarring).

2) How the state of IS is determined is not very clear. The duration value provided is given

with a precision that seems exaggerated compared to the indications in the raw data. The onset of IS sometimes coincides with a period with still little calcium activity, sometimes in the middle of a calcium peak (Fig 2d). According to Gottesmann, theta bouts start appearing during IS. Perhaps this increase in theta power could be seen in the parietal EEG? If this is not possible, strict duration indications should be replaced by approximate ranges.

First, we would like point out that our state scoring was done blindly with respect to the Ca^{2+} signal so the start and end of behavioural states, including IS, is not expected to match the specific dynamic of Ca^{2+} fluctuations. This unbiased approach actually strengthens the significance of our results.

The parameters used to define IS are explained in detail in the Methods section and represent standard characteristics described in the early work of Gottesmann and others (cited in the main text and Methods and see response to Reviewer#2/question 2 for a list of references). To characterise the IS, we use its three main hallmarks (all described in the Methods): increased in sigma power, increased in theta activity in the parietal EEG (as the reviewer points out) and decreased in delta activity. A summary of power changes in IS compared to NREM sleep is now reported as supplementary information (new Supplementary Figure 1b) and validates our scoring system. Combined, these criteria isolate stretches of IS episodes with an average duration of 44.43 ± 1.48 s ($n = 28$ rats, see Main text). This duration is in agreement with previous described values (~ 30 -s; (Gottesmann, 1992; Trachsel et al., 1988)).

3) The behavioral state classifications are quite arbitrary and do not follow standard rules. Microarousals can be shorter than 5 epochs and they will invariably disturb a consolidated bout. Moreover, they are frequent, in particular during head-restrained conditions. Therefore, classifying fragmented sleep into consolidated sleep bears the risk of inducing power fluctuations that can then affect the apparent time course of power values across apparently consolidated NREM sleep bouts.

Our state classification was done in 4-s epochs and follows strict rules using clearly defined EEG characteristics for each behavioural state. We agree with the reviewer that microarousals can be very brief and we observed sequences of wake epochs lasting < 20 -s (1 to 5 epochs) during NREM sleep episodes. We would like to point out that for Figures 3a, 5c, 6d we did not represent those short wake periods for clarity, which may have led to confusion about our waking state classification. We have modified the PD time courses in Figure 3 and added arrows to point at examples of brief microarousals during NREM sleep episodes (note their correspondence with decreases in sigma/ Ca^{2+} power). Then, our calculations of power fluctuations were done in 4-s epoch by normalising the mean power of each frequency band across the entire recording (i.e. irrespective of state classification/consolidation) (described in Methods). These power fluctuations were then matched with our state classification for each epoch and used for further analysis. It follows that the resulting time course of power fluctuations is entirely independent of vigilance state and on whether a NREM sleep bout is considered consolidated or not. Importantly, all of our correlation analysis (for freely behaving and head-fixed recordings) was performed between normalised power values of individual epochs for each state and did not consider state consolidation status. Therefore any “arbitrary” criteria of fragmented vs. consolidated state did not influence our main results.

4) It is not clear how NREM sigma power relates to the calcium PD during REM sleep in 3c.

At what moment during NREM sleep was sigma power calculated? Talking about sigma power during REM sleep is likely going to be very confusing and should be put into the supplemental materials.

We understand that sigma power calculations during REM sleep is not seen as traditional and could be confusing to the readers. We have removed the results related to sigma PD during REM sleep from the main Figure 3, but report the average correlation strength as numbers in the main text. Although surprising, this results is quite robust across animals and we feel that this information could be of interest for some readers, in particular for researchers in the sleep & developmental field. During early development, spindle-bursts occur regardless of sleep stages (Ellingson, 1982; Tiriac and Blumberg, 2016). A recent hypothesis even suggests that more sensitive detection methods might be useful to reveal the persistence of spindles during REM sleep into adulthood (Tiriac and Blumberg, 2016).

5) It could be assessed whether or not exits to different vigilance states from IS were in any way different in terms of IS parameters.

This is a very interesting point and we have performed a more detailed analysis of the IS and Ca^{2+} characteristics at transitions to wake, NREM or REM sleep to address this question. Results are reported in our new Supplementary Figure 5 and shows that neither IS oscillatory parameters nor Ca^{2+} activity in dendrites or L2/3 neurons show distinctive changes related to specific transitions. This result further supports our claim that IS may not be a transition state as previously described and represents part of NREM sleep's dynamic in rodents.

6) Ca signals were analyzed with respect to their frequency compositions using FFT. it is not clear how the data presented in 2b were obtained. First, the data suggest that the 0.1-1Hz component in the calcium signals in the dendrites is present in all vigilance states. However AW, QW, and also REM signals seem not to contain much of this (see 2a and 2d). Could the peak obtained by an artefact of doing the FFT over too short stretches of the behavioral states? Please provide the distribution of vigilance state bout durations and show that the FFT amplitude does not depend on the bout lengths included. Second, how was the FFT constructed given that its resolution will vary depending on bout length? Third, what do the individual data points represent in 2b and how was this binning done? Fourth, how are statistics done over a number of datapoints in the FFT that exceeds the number of animals used?

EEG and Ca^{2+} signal data processing is described in detail in the Methods section. The sampling rate of data acquisition was 200Hz for both Ca^{2+} and EEG signals and the FFT resolution 1024 data point (0.19Hz binning). The FFT power spectra for Ca^{2+} (and EEG) signals were computed in individual 4-s epochs for each behavioural state, which is a standard method used for more than 25 years for humans and animals EEG analysis. The resultant FFT power spectra are therefore by definition independent of vigilance state duration and represents the average FFT amplitude across all 4-s epochs for each behavioural state. Furthermore, the differences in vigilance state amounts and bout length (as requested by the reviewer, new graph in Fig. 1g), which are similar between groups, cannot explain the changes in the Ca^{2+} FFT amplitude in the slow component (<1Hz) in each state and between groups. The statistical comparisons of FFT datapoints between groups were performed using a two-

way ANOVA with “Group” and “Frequency” as variables followed by post-hoc comparisons for individual FFT datapoints between groups, similar to our previous published work (Aton et al., 2009a, 2009b; Bridi et al., 2015; Seibt et al., 2012).

7) The parallel increase of delta and sigma power during a NREM sleep bout is very surprising. It contradicts the widely documented notion that delta and sigma are anticorrelated during sleep. One reason could be that sleep was not measured at the same times of day, which could result in widely different delta power values. As also sleep bouts shorten with time of days, this could additionally lead to apparent increases. One idea would be to restrict this analysis to only certain periods of the day and to normalize the data only for each vigilance state.

We think that the reviewer is referring here to studies, mainly in humans, that describe an inverse dynamic of delta/SWA and sigma/spindles when averaged over extended periods (across the night in humans (Aeschbach and Borbély, 1993; Dijk et al., 1993) and light/dark phases in rodents (Trachsel et al., 1988; Vyazovskiy et al., 2004) or after sleep deprivation ((De Gennaro et al., 2000; Dijk et al., 1993; Schwierin et al., 1999)). Our study does not investigate this time scale of dynamics. Assuming that this delta/sigma anticorrelation trend is similar over shorter sleep periods/within sleep episodes, as we investigate here, is an oversimplification. In humans, SWA/delta and sigma/spindles rise and decline in parallel at the beginning and end of an NREM episodes, respectively (Aeschbach and Borbély, 1993; Dijk et al., 1993). Recent studies in rodents support our current findings showing a gradual and parallel increase of a wide range of frequency bands (including delta & sigma, see Supplementary Fig.7) in the hippocampus and cortex of rats during individual NREM episodes (that last often < 10 min.) (Grosmark et al., 2012; Watson et al., 2016). In addition, states amounts, bout durations and “time of day” were the same across groups (Fig. 1c,g) justifying our normalisation approach across behavioural states. Our results are therefore not surprising and reflect power fluctuations within vigilance states rather than over the 24-h day cycle, which show different dynamics.

8) On top of the 0.1-1Hz fluctuations, the data in Fig3 suggest that there is an additional infra-slow fluctuation over which both Ca and sigma PD vary. This is strikingly reminiscent of a report recently published (Lecci et al., Science Advances, February 2017) that reports on a 0.02-Hz oscillation in sigma power that is anticorrelated with heart rate and that determines behavioral arousability. It would be interesting to determine whether the 0.02 Hz-frequency component is also present in this study.

We thank the reviewer for pointing that out. Following the publication by Lecci and collaborators, we were very curious to investigate this aspect in our own data. Since our analysis of power (PD) time course uses smoothed PD values (moving average, see Methods), we performed the infra-slow FFT analysis on both the raw (similar to the Lecci et al. study) and the smoothed sigma power fluctuations. Both type of data gave similar results and revealed a slow component that, on average, displays a peak at slower frequencies than 0.02Hz. This peak was clearer and more pronounced for smoothed PD values and also present for dendritic Ca²⁺ fluctuations (Figure IIIa). The shift in peak towards slower frequencies is not due to methodological reasons as the data were processed the same way as in the Lecci et al. study and analysed by both laboratories (i.e. Lüthi’s and ours) in parallel. Among the NREM episodes, some did show a distinct peak of sigma power at 0.02Hz (episodes #1 & #2,

Fig. IIIb) but others displayed a peak at slower frequency ($\sim 0.01\text{Hz}$, episodes #3 & #4, Fig. IIIb). This slower component around $0.01\text{-}0.015\text{Hz}$ was quite frequent, especially for longer episodes, and might reflect the recently described “NREM packets” within NREM episodes in rats (Watson et al., 2016). This distribution of infra-slow oscillations peaks between $0.01\text{-}0.02\text{Hz}$ also explains that, on average, we obtain a slower power frequency peak in our results. Since the main difference between our data and the Lecci et al. study is the rodent species (mice vs. rats), it is possible that rats have a slightly different infra-slow dynamic (e.g. organisation into NREM packets in rats). We decided to not include these data in the paper as

Figure III. Infra-slow analysis (a) Example of a NREM episode (~ 13 min) used for infra-slow FFT analysis that shows the raw (blue; data format used in the Lecci et al. study) and smoothed (black, see Methods) sigma PD time course (normalized to the mean across behavioural states). The corresponding average mean power (red lines) spectra ($n = 22$ rats [L2/3 & Dendrites], grey lines) in the infra-slow band are represented on the right. The inset graph represents the same analysis for smoothed Ca^{2+} power fluctuations in dendrites ($n = 11$) (b) Raw power traces (black line) and corresponding power spectra (red) for representative examples of sigma fluctuations during NREM episodes showing peaks at 0.02Hz (episodes #1 & #2) and at slower frequencies (episodes #3 & #4). The duration of each NREM episode is represented on the x-axis (in seconds).

we feel that a more controlled and systematic comparisons between rodent species is needed to fully address the question. However, we could consider to include the figure as Supplementary material as per request.

9) The authors should apply some caution in equating sigma power with spindles. It would need to be shown directly that increased sigma power goes along with elevated spindle density, and spindles would need to be counted.

We thank the reviewer for raising this important question. We took advantage of our combined EEG/LFP recordings to investigate this question. The new results are presented in Figure 7 (new panel f). For this analysis we assessed the correlation strength between spindle density and EEG/LFP power for all frequency bands used in this study. We confirm that increased sigma power (in both EEG and LFP) is a specific predictor of increased spindle density compared with other frequency bands. To our surprise, beta oscillations, at least in rats, were also specifically correlated with spindle density (Fig. 7f). This result has two important consequences. Firstly, it suggests that the nature of cortical and thalamic spindles, while linked mechanistically, might be different. Secondly, and importantly, this relationship reinforced the significance of our results as they suggest that our correlations between dendritic activity and sigma-beta oscillations is a direct reflection of a correlation with cortical spindle events. Our results thus demonstrate that EEG sigma power does reflect spindle events and further suggest that a larger frequency band in the sigma-beta range can be used as EEG hallmark of cortical spindles in rats.

10) Any idea of how the 9-16 Hz rhythmic inputs from thalamus are transformed into 0.1-1 Hz calcium oscillations? In the non-filtered calcium signal, is there any evidence for faster phasic events that could reflect individual spindles? Alternatively, should we start thinking that spindle-related thalamocortical EPSPs are temporally integrated such that the original spindle frequency disappears and that dendrites become disinhibited over a time scale 1-2 times slower than the spindle events? could this be a reason that there is still much controversy going on about how to define cortical spindles?

We are happy to speculate on this question. We know that dendritic Ca^{2+} is exquisitely sensitive to dendritic inhibition with specialized inhibitory circuits leading to both GABA_A and GABA_B inhibition (Palmer et al., 2012; Pérez-García et al., 2006). The GABA_A inhibition lasts for up to 150 ms and the GABA_B for up to 500 ms. If the spindles lead to a disinhibition of these circuits, it could lead to a conversion to these frequencies (i.e. around 1 Hz). On the other hand, although we currently don't know the exact source of the dendritic Ca^{2+} , we do know that the dendrites of L5 pyramidal neurons are capable of long, plateau-like Ca^{2+} events (both Ca^{2+} and NMDA spikes) that can last for up to 100 ms with a long refractory period, so that dendritic Ca^{2+} events might express an intrinsic slow oscillatory frequency. Finally, the intrinsic frequency of occurrence of spindles (every ~5 seconds; 0.21 ± 0.02 Hz, $n = 9$ recordings rich in spindles) may be an underlying factor. In any case, we found no obvious fast oscillatory activity in the Ca^{2+} power spectra that could account for individual spindle events.

11) Why was the 2P imaging not done over hindlimb area in mice?

We were also interested to test the generalizability of the initial results in our study and so chose to do the two-photon imaging in an adjacent region of the somatosensory cortex. Since the results were consistent in both regions, we assume that the exact region chosen was not a factor. We chose not to stress this in the writing of the paper because it would require a larger study and probably multiple regions.

References:

- Aeschbach, A., and Borbély, A.A. (1993). All-night dynamics of the human sleep EEG. *J. Sleep Res.* 2, 70–81.
- Aton, S.J., Seibt, J., Dumoulin, M., Jha, S.K., Steinmetz, N., Coleman, T., Naidoo, N., and Frank, M.G. (2009a). Mechanisms of sleep-dependent consolidation of cortical plasticity. *Neuron* 61, 454–466.
- Aton, S.J., Seibt, J., Dumoulin, M.C., Coleman, T., Shiraishi, M., and Frank, M.G. (2009b). The sedating antidepressant trazodone impairs sleep-dependent cortical plasticity. *PLoS One* 4, e6078.
- Bridi, M.C.D., Aton, S.J., Seibt, J., Renouard, L., Coleman, T., and Frank, M.G. (2015). Rapid eye movement sleep promotes cortical plasticity in the developing brain. *Sci. Adv.* 1, e1500105.
- De Gennaro, L., Ferrara, M., and Bertini, M. (2000). Topographical distribution of spindles: variations between and within nrem sleep cycles. *Sleep Res. Online SRO* 3, 155–160.
- Dijk, D.J., Hayes, B., and Czeisler, C.A. (1993). Dynamics of electroencephalographic sleep spindles and slow wave activity in men: effect of sleep deprivation. *Brain Res.* 626, 190–199.
- Ellingson, R.J. (1982). Development of sleep spindle bursts during the first year of life. *Sleep* 5, 39–46.
- Gottesmann, C. (1992). Detection of seven sleep-waking stages in the rat. *Neurosci. Biobehav. Rev.* 16, 31–38.
- Grosmark, A.D., Mizuseki, K., Pastalkova, E., Diba, K., and Buzsáki, G. (2012). REM sleep reorganizes hippocampal excitability. *Neuron* 75, 1001–1007.
- Palmer, L.M., Schulz, J.M., Murphy, S.C., Ledergerber, D., Murayama, M., and Larkum, M.E. (2012). The cellular basis of GABA(B)-mediated interhemispheric inhibition. *Science* 335, 989–993.
- Pérez-Garci, E., Gassmann, M., Bettler, B., and Larkum, M.E. (2006). The GABAB1b isoform mediates long-lasting inhibition of dendritic Ca²⁺ spikes in layer 5 somatosensory pyramidal neurons. *Neuron* 50, 603–616.
- Schwierin, B., Achermann, P., Deboer, T., Oleksenko, A., Borbély, A.A., and Tobler, I. (1999). Regional differences in the dynamics of the cortical EEG in the rat after sleep deprivation. *Clin. Neurophysiol. Off. J. Int. Fed. Clin. Neurophysiol.* 110, 869–875.
- Seibt, J., Dumoulin, M.C., Aton, S.J., Coleman, T., Watson, A., Naidoo, N., and Frank, M.G. (2012). Protein Synthesis during Sleep Consolidates Cortical Plasticity In Vivo. *Curr. Biol. CB* 22, 676–682.
- Tiriác, A., and Blumberg, M.S. (2016). The Case of the Disappearing Spindle Burst. *Neural Plast.* 2016, 8037321.
- Trachsel, L., Tobler, I., and Borbély, A.A. (1988). Electroencephalogram analysis of non-rapid eye movement sleep in rats. *Am. J. Physiol.* 255, R27–37.
- Vyazovskiy, V.V., Achermann, P., Borbély, A.A., and Tobler, I. (2004). The dynamics of spindles and EEG slow-wave activity in NREM sleep in mice. *Arch. Ital. Biol.* 142, 511–523.
- Watson, B.O., Levenstein, D., Greene, J.P., Gelinas, J.N., and Buzsáki, G. (2016). Network Homeostasis and State Dynamics of Neocortical Sleep. *Neuron*.

REVIEWERS' COMMENTS:

Reviewer #1 (Remarks to the Author):

The authors have satisfied my concerns. In fact, I think the authors have done a commendable job of addressing the concerns of all 3 reviewers. This is an elegant and exciting paper and I look forward to follow-on studies from this lab that will extend this important foundational work by investigating the causal relationship between dendritic Ca²⁺ activity, spindles and cortical plasticity.

Reviewer #2 (Remarks to the Author):

The authors have sufficiently addressed all questions raised by myself and the other reviewers. I am satisfied and see no reason to delay publication.

Reviewer #3 (Remarks to the Author):

The authors have done a great job in revising this paper that will be a very welcome study in the sleep field.

I have two minor comments:

- please be more specific in the methods explaining how individual spindles were detected for the data in Figure 7. How were onset and offset times determined? In the axes labeled with spindle density, it would be helpful to specify that density is plotted as a cumulative duration, as one expects a per-time unit.

Why was cumulative duration and not discrete spindles number plotted against sigma power?

- check spelling for theta (not tehta) throughout ms and suppl file

Congratulations on this work! Anita Luthi

REVIEWERS' COMMENTS:

We are pleased that we could address all the concerns raised by reviewer#1 and reviewer#2. We also thank again all the reviewers for their helpful suggestions and comments.

Reviewer #1 (Remarks to the Author):

The authors have satisfied my concerns. In fact, I think the authors have done a commendable job of addressing the concerns of all 3 reviewers. This is an elegant and exciting paper and I look forward to follow-on studies from this lab that will extend this important foundational work by investigating the causal relationship between dendritic Ca²⁺ activity, spindles and cortical plasticity.

Thank you, no response needed.

Reviewer #2 (Remarks to the Author):

The authors have sufficiently addressed all questions raised by myself and the other reviewers. I am satisfied and see no reason to delay publication.

Thank you, no response needed.

Reviewer #3 (Remarks to the Author):

The authors have done a great job in revising this paper that will be a very welcome study in the sleep field.

Thank you.

I have two minor comments:

- please be more specific in the methods explaining how individual spindles were detected for the data in Figure 7. How were onset and offset times determined?

We used visual detection of spindles which are easily identifiable in the LFP (unlike in the EEG). The onset of a spindle, which often followed an UP state, was determined as the beginning of the train of oscillations (frequency between 10.4Hz to 19.9Hz; duration between 0.5-1.8 seconds; N = 100 representative spindles [50/animal]). The offset was determined by the termination of this train of oscillations, usually characterised by the onset of another UP state (or another state). We have added this information in the Methods.

In the axes labeled with spindle density, it would be helpful to specify that density is plotted as a cumulative duration, as one expects a per-time unit.

We agree and changed the x-axis label.

Why was cumulative duration and not discrete spindles number plotted against sigma power?

We thank the reviewer for pointing that out. Our reasoning is that spindle number does not reflect spindle duration which also contributes to density (i.e. spindle activity is doubled for two spindles of 2s compared to two spindles of 1s within the same time window). We opted

for the spindle cumulative duration that takes into account both the number and the duration of spindles and thus, we think, reflects more accurately spindle density. This is particularly important when compared to EEG/LFP power changes that also vary with oscillatory activity duration.

- check spelling for theta (not tehta) throughout ms and suppl file

The spelling mistake has been checked and corrected in all files.

Congratulations on this work! Anita Luthi